# ALL PATCHES MATTER, MORE PATCHES BETTER: ENHANCE AI-GENERATED IMAGE DETECTION VIA PANOPTIC PATCH LEARNING

**Zheng Yang**[1][*], **Ruoxin Chen**[2][*][†], **Zhiyuan Yan**[3], **Keyue Zhang**[2], **Xinghe Fu**[1], **Shuang Wu**[2], **Xiujun Shu**[4], **Taiping Yao**[2], **Shouhong Ding**[2], **Zequn Qin**[5][†], **Xi Li**[1][†]

[1]College of Computer Science and Technology, Zhejiang University
[2]Youtu Lab, Tencent     [3]Peking University     [4]Wechat Pay, Tencent
[5]School of Software Technology, Zhejiang University

## ABSTRACT

The rapid proliferation of AI-generated images (AIGIs) highlights the pressing demand for generalizable detection methods. In this paper, we establish two key principles for AIGI detection task through systematic analysis: **(1) All Patches Matter**, since the uniform generation process ensures that each patch inherently contains synthetic artifacts, making every patch a valuable detection source; and **(2) More Patches Better**, as leveraging distributed artifacts across more patches improves robustness by reducing over-reliance on specific regions. However, counterfactual analysis uncovers a critical weakness: naively trained detectors display **Few-Patch Bias**, relying disproportionately on *minority patches*. We identify this bias to **Lazy Learner** effect, where detectors to limited patch artifacts while neglecting distributed cues. To address this, we propose **Panoptic Patch Learning** framework, which integrates: (1) *Randomized Patch Reconstruction*, injecting synthetic cues into randomly selected patches to diversify artifact recognition; (2) *Patch-wise Contrastive Learning*, enforcing consistent discriminative capability across patches to ensure their uniform utilization. Extensive experiments demonstrate that PPL enhances generalization and robustness across datasets.

## 1 INTRODUCTION

The rapid evolution of generative AI models has precipitated an exponential growth of AI-generated images (AIGIs) in digital ecosystems (Goodfellow et al., 2014; Karras et al., 2018; 2019; Ho et al., 2020; Rombach et al., 2022; Zhang et al., 2023; Ramesh et al., 2021; Yan et al., 2025b; 2024c; 2025a; Zhang et al., 2025; 2024; Song et al., 2025b;a). This proliferation raises concerns regarding information security and content authenticity, highlighting the need for AIGI detection to distinguish synthetic images from authentic ones. Unlike conventional classification tasks, AIGI detection operates as a "*cat-and-mouse game*", presenting unique challenges due to: (1) continuous emergence of new generative architectures, and (2) frequent updates to existing generative models. Consequently, exhaustive training on all synthetic data becomes impractical (Ojha et al., 2023), thus necessitating detectors with strong generalizability.

Despite these challenges, AIGIs exhibit a distinctive property absent in traditional classification: Universal Artifact Distribution. In the context of AIGIs, discriminative features are not confined to object-centric regions; instead, *synthetic images contain artifacts uniformly across all patches, a consequence of the consistent generation process of modern generative models.*[1] This indicates that every patch, defined as partitioned local sub-blocks, contains synthetic traces, forming our first principle for AIGI detection: All Patches Matter. This principle is supported by two lines of evidence: (1) visual analytics (Tan et al., 2024c; Cozzolino et al., 2024; Chen et al., 2025) confirm pixel-level discriminative patterns, revealing artifacts at patch granularity; and (2) recent patch-wise detectors (Chen et al., 2024b; Zhong et al., 2024) demonstrate comparable performance to full-image

---

[*]: Equal Contribution, [†]: Corresponding Author

[1]This work adheres to the mainstream AIGI detection setting (Chen et al., 2024a; Ojha et al., 2023; Tan et al., 2024a; Liu et al., 2024; He et al., 2024; Zhu et al., 2024; Tan et al., 2024c; Lin et al., 2025) where the entire image is generated by AI models.

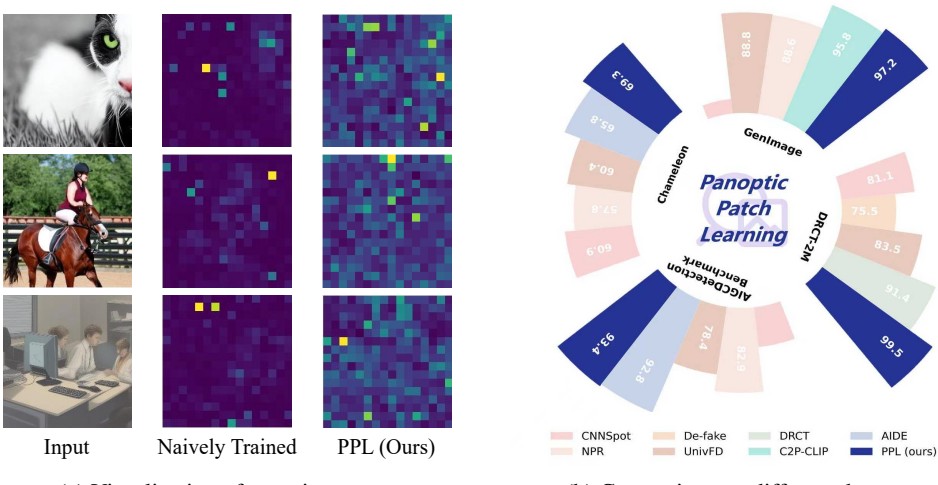

(a) Visualization of attention maps.      (b) Comparison on different datasets.

Figure 1: **(a)** PPL produces a more uniform attention distribution across patches, indicating its effectiveness in capturing artifacts comprehensively. **(b)** PPL outperforms peer methods on GenImage, DRCT-2M, AIGCDetectionBenchmark, and the in-the-wild Chameleon. More details are in Section 5.

approaches, validating the discriminative capability of individual patches. Although artifacts vary across patches, detectors that capture diverse artifacts across distributed regions reduce over-reliance on specific patches. Capturing these distributed artifacts enhances cross-generator generalizability by mitigating detectors' blind spots. This leads to our second principle: More Patches Better.

However, counterfactual analysis of existing AIGI detectors (Ojha et al., 2023; Liu et al., 2024; Tan et al., 2024a; He et al., 2024; Chen et al., 2024a) reveals an unfavorable tendency—Few-Patch Bias—supported by two empirical observations and a quantitative analysis. Empirically, we observe: (1) detectors' attention maps disproportionately focus on very limited patches; (2) detectors exhibit severe patch-specific fragility, where masking a single patch reduces accuracy by $18.7\% \pm 4.1\%$ on average. Quantitatively, using causal inference tool CDE (VanderWeele, 2013) to quantify each patch's impact—measured as the classification logit difference with and without that patch—we find that naively trained detectors produce skewed distributions: a few patches show high CDE values, while most patches exhibit significantly lower contributions. This suggests that most patches remain underutilized, despite also containing generative artifacts. Moreover, *detection methods with more uniform CDE distributions exhibit stronger generalizability*; for instance, DRCT, with more high-CDE patches, performs substantially better than UnivFD. We attribute such *Few-Patch Bias* to the propensity of detectors as *Lazy Learner* (Hermann et al., 2024; Zhang et al., 2021; Wang et al., 2022; Zhao et al., 2024; Ghosh et al., 2023; Tang et al., 2023; Sun et al., 2024; Yuan et al., 2024; Yan et al., 2024b). Specifically, AIGI detectors follow a curriculum-like learning pattern: *once easy-to-learn artifacts in certain patches minimize loss, the presence of these patches discourages exploration of broader regions.*

To address this challenge, we propose the principle: "All Patches Matter, More Patches Better", which prevents detectors from shortcutting to a few regions and instead encourages robust feature learning across the entire image. To operationalize this principle, we introduce the **Panoptic Patch Learning** (PPL) framework, which consists of two components: (1) *Randomized Patch Reconstruction*, which manually injects synthetic artifacts into randomly selected patches of real images via diffusion reconstruction, forcing the model to discriminate based on these chosen regions and discouraging over-reliance on specific patches; and (2) *Patch-wise Contrastive Learning*, which aligns the representations of real and synthetic patches, thereby enforcing consistent discriminative capability across all regions of the image. Fig. 1 illustrates the effectiveness of PPL. Our main contributions are threefold:

1. We formally propose the principle "All Patches Matter, More Patches Better", showing that exploiting distributed artifacts enhances AIGI detection.

2. We provide a detailed patch-wise analysis using CDE, revealing that Few-Patch Bias is pervasive in existing detectors.

3. Building on this principle, we design Panoptic Patch Learning and validate its effectiveness through extensive experiments.

## 2 RELATED WORK

Existing AIGI detection methods can be broadly categorized into two types: *local* and *global* detection (Tan et al., 2024a). We summarize both lines of research below.

**Local AIGI detection methods.** Local approaches exploit localized information to distinguish AI-generated images from real ones, assuming that low-level feature differences exist between the two. These methods can be divided into *patch-wise* and *pixel-wise* detectors.

*Patch-wise methods* include: SSP (Chen et al., 2024b) achieves notable performance using only a single patch. Patchcraft (Zhong et al., 2024) separates processing of the simplest and most complex patches by entropy-based selection. (Zheng et al., 2024) employ a patch-based CNN leveraging all patches to avoid selective sampling and aggregate patch features. TextureCrop (Konstantinidou et al., 2025) partitions an image via sliding windows and selects high-frequency texture-rich regions. Despite these advances, patch-wise detectors often over-rely on a limited subset of patches, leading to information under-utilization. *Pixel-wise methods* include: NPR (Tan et al., 2024c) detects AIGIs by analyzing differences in neighboring pixel relationships. FreqNet (Tan et al., 2024b) and SAFE (Li et al., 2024) exploit high-frequency signals to capture localized patterns. However, pixel-wise methods are sensitive to small perturbations in pixel relationships, limiting their robustness.

**Global AIGI detection methods.** Global approaches leverage holistic image characteristics to distinguish AIGIs from real images, aiming to capture inconsistencies that may not be observable at the local level. CNNSpot (Wang et al., 2020) applies a CNN directly for detection, achieving strong in-distribution performance but suffering from poor cross-generator generalization. UnivFD (Ojha et al., 2023) improves robustness by adopting a CLIP visual encoder as a feature extractor. FatFormer (Liu et al., 2024) further adapts CLIP by introducing a frequency adapter. C2P-CLIP (Tan et al., 2024a) fine-tunes CLIP with carefully designed image–text pairs to embed the notions of "real" and "fake." DRCT (Chen et al., 2024a) strengthens UnivFD with a contrastive loss on hard cases. Nevertheless, global methods often overlook fine-grained forensic artifacts, which constrains their effectiveness.

## 3 MOTIVATION

### 3.1 ALL PATCHES MATTER, MORE PATCHES BETTER

The principle of *All Patches Matter* is supported by three key findings.

1. **Principle:** Because every patch of a synthetic image is itself generated, each inherently contains artifacts. Localized detection methods (Chen et al., 2024b; Zhong et al., 2024) demonstrate that cues within small regions can effectively discriminate real from synthetic content, underscoring that every patch carries discriminative signals.

2. **Visualization:** Fig. 2 illustrates distinct artifact patterns across patches, showing that each synthetic patch exhibits identifiable features distinguishing it from real patches. Moreover, the variability of these cues across patches highlights the considerable diversity of artifacts present in synthetic images.

3. **Experiments:** We further validated this principle by evaluating detectors on single randomly selected patches. By replicating one patch across the image to isolate its features, detectors still achieved 90% accuracy on the SDv1.4 subset of GenImage. This confirms that even a single patch contains sufficient information for reliable discrimination.

Together, these findings demonstrate that artifacts in synthetic images are both pervasive and diverse. Detectors can exploit these patch-level cues, motivating the principle of *More Patches Better*: leveraging more patches enhances robustness and generalization by capturing complementary artifact patterns. However, our observations reveal that existing detectors do not align with this principle.

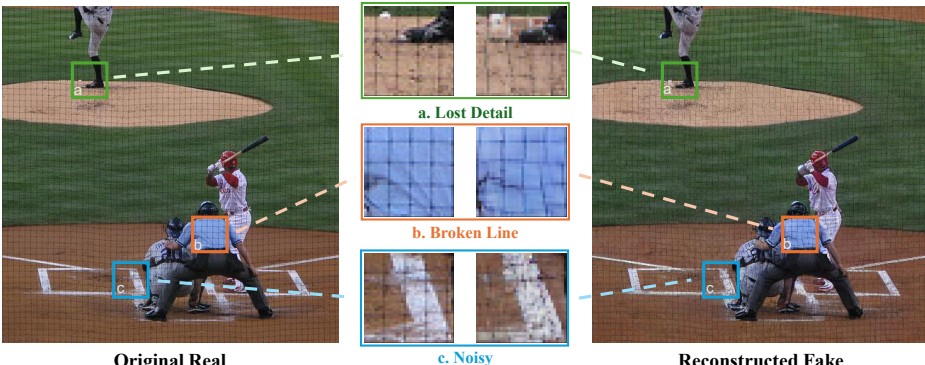

Figure 2: Visual evidence of patch-wise artifacts. We observe diverse traces—such as broken lines, unnatural noise, and boundary detail loss—showing that multiple regions of synthetic images contain cues. This observation underscores the importance of *leveraging more patches to enhance recognition of diverse artifacts*. Images are sourced from MSCOCO (Lin et al., 2014).

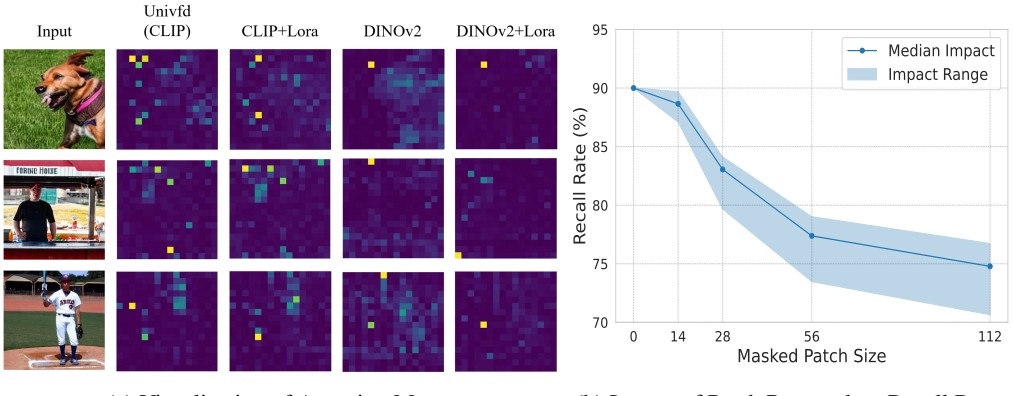

(a) Visualization of Attention Maps.    (b) Impact of Patch Removal on Recall Rate.

Figure 3: **(a)** Attention maps reveal the *few-patch bias* of naively trained detectors, where attention concentrates on a small number of dominant patches, reflecting over-reliance on limited regions. **(b)** Recall degradation occurs when single patches of varying sizes are occluded, showing that detectors are overly sensitive to corruption in specific regions and suffer notable performance drops.

## 3.2   FEW-PATCH BIAS

**Observations.**   Our empirical observations indicate that existing detectors often overly rely on a limited number of patches. Our experiments reveal that existing detectors tend to over-rely on a limited set of patches. Fig. 3(a) shows attention maps from naively trained ViTs, where attention weights concentrate on only a few regions. This phenomenon persists even when changing the backbone or applying LoRA, suggesting a model-agnostic bias. To further validate this observation, we systematically mask patches of varying sizes and measured the corresponding recall rate degradation. Fig. 3(b) illustrates the performance of UnivFD under patch occlusion. Masking a single patch leads to a substantial drop in accuracy, and the impact varies across different patches, confirming that detectors are disproportionately sensitive to specific regions.

**Quantitative analysis.**   Building on the above observations, we employ the Controlled Direct Effect (CDE) to quantify the impact of each patch. Conceptually, if both $X \rightarrow Y$ and $Z \rightarrow Y$, then the outcome $Y$ results from the combined influence of $X$ and $Z$. The CDE measures the contribution of $X$ by comparing outcomes with and without its effect while keeping other factors fixed.

For an image, the CDE of the patch at row $i$, column $j$ is defined as:

$$CDE := \delta_I - \delta_{I-(i,j)}, \quad \delta := \text{logit}_{synth} - \text{logit}_{real}, \tag{1}$$

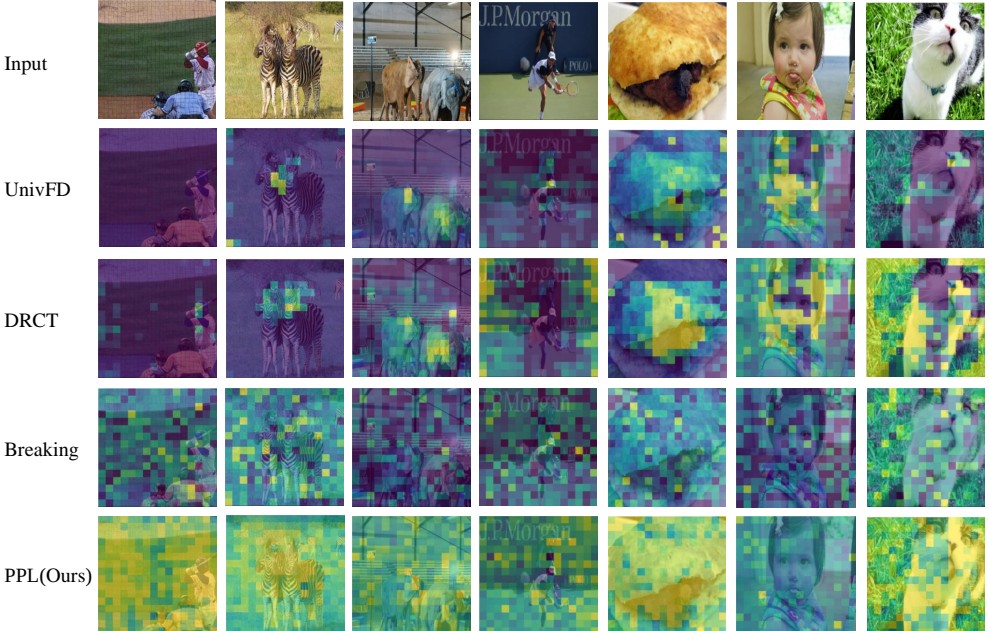

Figure 4: CDE heatmap of existing methods on generated images selected from the DRCT-2M dataset (Chen et al., 2024a). A broader and more uniform highlighted region indicates a greater number of patches contributing to determining a fake image.

where $I$ denotes the original image and $I - (i, j)$ the image with the $(i, j)$-th patch masked (implemented by setting the patch values to zero). By computing the CDE for each patch, we quantify its relative contribution to the synthetic classification decision.

Fig. 4 presents CDE heatmaps. From top to bottom, the number of active patches increases and the CDE distributions become more uniform. Stronger detectors consistently activate a broader set of patches. These visualizations highlight the prevailing bias toward a few dominant patches with disproportionately high CDE, motivating us to mitigate few-patch reliance.

## 4 METHODOLOGY

Panoptic Patch Learning is a comprehensive framework based on the principles of "All Patches Matter" and "More Patches Better," achieved through innovative data and learning strategies, as illustrated in Fig. 5. Specifically, the data strategy, Randomized Patch Reconstruction (RPR), discourages the model from over-relying on any specific patches, thereby enhancing its recognition capability for various artifacts across *more patches*. Following this, the learning strategy, Patch-wise Contrastive Learning (PCL), ensures that *all patches*, both frequently attended and underutilized, are brought closer in the feature space, thereby uniformizing the impact of all patches.

**Randomized Patch Reconstruction encourages "More Patches Better".** The RPR process is carried out by performing diffusion reconstruction on randomly selected patches with a specified proportion, injecting synthetic cues into specific regions of the image while maintaining the overall semantics of the image (as the reconstructed image closely resembles the original image). In practice, RPR is implemented by first applying diffusion reconstruction to the entire image to obtain a reconstructed version. Then, the selected patches in the original image are replaced with their reconstructed counterparts, resulting in a synthetic image where only specific regions contain synthetic artifacts. Here, we emphasize that we inject synthetic features via diffusion reconstruction rather than stitching a synthetic patch, in order to preserve the global semantics and integration of the produced image, and to prevent the model from overfitting to images with disconnected semantics. We use $r \in [0, 1.0]$ to denote the ratio of reconstructed patches relative to the whole image.

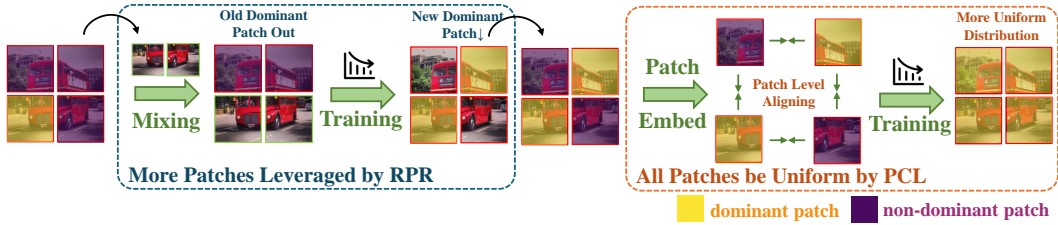

Figure 5: The Panoptic Patch Learning (PPL) framework embodies the principles of **All Patches Matter** and **More Patches Better** through two key components: Randomized Patch Reconstruction (RPR) and Patch-wise Contrastive Learning (PCL). During training, the model may excessively rely on **dominant patches**, neglecting others. RPR mitigates this by randomly replacing dominant patches with real ones, encouraging the model to detect artifacts in **non-dominant patches** and thereby expanding the coverage of dominant regions. PCL further promotes balanced patch utilization by aligning the embeddings of patches with the same labels. Together, RPR and PCL foster comprehensive and uniform exploitation of patches.

---

**Algorithm 1** Patch-wise Contrastive Learning (PCL) Training Procedure

---

**Input:**

    $X$       – input image tensor after RPR, shape $[B, C, H, W]$

    $label_{gt}$    – image-level ground-truth labels, shape $[B]$

    $patch_{gt}$    – patch-level ground-truth labels, shape $[B, K]$

    $\lambda$       – weighting coefficient for contrastive loss

1: $img_{embedding}, patch_{embedding} \leftarrow ViT_{Encoder}(X) \triangleright img_{embedding}: [B, 1, D], patch_{embedding}:$ $[B, K, D]$

2: $y_{pred} \leftarrow \text{Linear}(img_{embedding})$            $\triangleright$ Image-level class logits, shape $[B, 2]$

3: $L_{ce} \leftarrow \text{BCELoss}(y_{pred}, label_{gt})$        $\triangleright$ Image-level classification loss

4: $L_{con} \leftarrow \text{ContrastiveLoss}(patch_{embedding}, patch_{gt})$    $\triangleright$ Patch-level contrastive loss

5: $L_{total} \leftarrow \lambda \cdot L_{con} + (1 - \lambda) \cdot L_{ce}$

6: $L_{total}.\text{backward}()$

---

**Patch-wise Contrastive Learning emphasizes "All Patches Matter".** PCL operationalizes the principle of "All Patches Matter" by aligning the embedding vectors of different patches, bringing patches with identical labels closer together while distancing those with different labels. We employ contrastive learning to cluster synthetic patches more closely within each batch, while maintaining a margin that separates synthetic and real patches. This approach ensures that if an image contains any dominant patch with easily learnable artifacts, the model improves its performance on the remaining patches, thus promoting the utilization of all patches. Specifically, for each batch, we utilize a margin-based contrastive loss (Hadsell et al., 2006):

$$\mathcal{L}_{con} = \sum_{i,j:\ i \neq j} \left[ Y \cdot d^2 + (1 - Y) \cdot \max\left(0, \alpha - d^2\right) \right], \tag{2}$$

where $i, j$ represent the indices of patch tokens within a batch. $d$ measures the Euclidean distance between the patch embeddings. $\alpha$ defines a minimum distance threshold between negative sample pairs, thereby enhancing the model's ability to distinguish between similar and dissimilar pairs. $Y$ indicates whether two patches share identical labels, thus pulling positive patch pairs closer and pushing negative patch pairs further apart. The overall loss function is a weighted combination of the cross-entropy loss and the patch-wise contrastive loss:

$$\mathcal{L}_{\text{total}} = \lambda \mathcal{L}_{\text{con}} + (1 - \lambda) \mathcal{L}_{\text{ce}}, \tag{3}$$

The practical implementation of PCL is shown in Alg. 1.

## 5 EXPERIMENTS

**Implementation details.** We adopt CLIP (Radford et al., 2021) and DINOv2 (Oquab et al., 2023) two vision foundation model as backbones , and fine-tune them using LoRA. Unless otherwise

Table 1: Cross-model accuracy (Acc) on GenImage. All methods are trained on the SDv1.4 subset. Results are taken from C2P-CLIP (Tan et al., 2024a), except SAFE and Effort, which are reported in their original papers. For Breaking (Zheng et al., 2024), we re-implement the method because no GenImage results or checkpoints are publicly available. Our results are bolded when they achieve the highest accuracy among all methods.

| Method | Ref | Midjourney | SDv1.4 | SDv1.5 | ADM | GLIDE | Wukong | VQDM | BigGAN | mAcc |
|---|---|---|---|---|---|---|---|---|---|---|
| ResNet-50 (He et al., 2016) | CVPR2016 | 54.9 | 99.9 | 99.7 | 53.5 | 61.9 | 98.2 | 56.6 | 52.0 | 72.1 ± 22.6 |
| DeiT-S (Touvron et al., 2021) | ICML2021 | 55.6 | 99.9 | 99.8 | 49.8 | 58.1 | 98.9 | 56.9 | 53.5 | 71.6 ± 23.2 |
| Swin-T (Liu et al., 2021) | ICCV2021 | 62.1 | 99.9 | 99.8 | 49.8 | 67.6 | 99.1 | 62.3 | 57.6 | 74.8 ± 21.1 |
| CNNSpot (Wang et al., 2020) | CVPR2020 | 52.8 | 96.3 | 95.9 | 50.1 | 39.8 | 78.6 | 53.4 | 46.8 | 64.2 ± 22.6 |
| Spec (Zhang et al., 2019) | WIFS2019 | 52.0 | 99.4 | 99.2 | 49.7 | 49.8 | 94.8 | 55.6 | 49.8 | 68.8 ± 24.1 |
| F3Net (Qian et al., 2020) | ECCV2020 | 50.1 | 99.9 | 99.9 | 49.9 | 50.0 | 99.9 | 49.9 | 49.9 | 68.7 ± 25.8 |
| GramNet (Liu et al., 2020) | CVPR2020 | 54.2 | 99.2 | 99.1 | 50.3 | 54.6 | 98.9 | 50.8 | 51.7 | 69.9 ± 24.2 |
| UnivFD (Ojha et al., 2023) | CVPR2023 | 93.9 | 96.4 | 96.2 | 71.9 | 85.4 | 94.3 | 81.6 | 90.5 | 88.8 ± 8.6 |
| NPR (Tan et al., 2024c) | CVPR2024 | 81.0 | 98.2 | 97.9 | 76.9 | 89.8 | 96.9 | 84.1 | 84.2 | 88.6 ± 8.3 |
| FreqNet (Tan et al., 2024b) | AAAI2024 | 89.6 | 98.8 | 98.6 | 66.8 | 86.5 | 97.3 | 75.8 | 81.4 | 86.8 ± 11.6 |
| FatFormer (Liu et al., 2024) | CVPR2024 | 92.7 | 100.0 | 99.9 | 75.9 | 88.0 | 99.9 | 98.8 | 55.8 | 88.9 ± 15.7 |
| DRCT (Chen et al., 2024a) | ICML2024 | 91.5 | 95.0 | 94.4 | 79.4 | 89.1 | 94.6 | 90.0 | 81.6 | 89.4 ± 5.9 |
| Effort (Yan et al., 2024b) | ICML2025 | 82.4 | 99.8 | 99.8 | 78.7 | 93.3 | 97.4 | 91.7 | 77.6 | 91.1 ± 11.8 |
| Breaking (Zheng et al., 2024) | NIPS2024 | 83.9 | 98.9 | 93.0 | 99.1 | 97.7 | 85.4 | 92.7 | 90.5 | 92.7 ± 5.8 |
| SAFE (Li et al., 2024) | KDD2025 | 95.3 | 99.4 | 99.3 | 82.1 | 96.3 | 98.2 | 96.3 | 97.8 | 95.6 ± 5.6 |
| C2P-CLIP (Tan et al., 2024a) | AAAI2025 | 88.2 | 90.9 | 97.9 | 96.4 | 99.0 | 98.8 | 96.5 | 98.7 | 95.8 ± 4.0 |
| Ours/DINOv2 | | 90.4 | 98.2 | 97.7 | 91.8 | 96.3 | 98.0 | 97.7 | 96.2 | **95.9** ± 3.0 |
| Ours/CLIP | | 94.8 | 98.5 | 98.3 | 94.7 | 96.1 | 98.6 | 98.5 | 98.0 | **97.2** ± 1.7 |

Table 2: Cross-model accuracy (Acc) on DRCT-2M. All methods are trained on the SDv1.4 subset. Results of other methods are taken from DRCT (Chen et al., 2024a).

| Method | SD Variants | | | | | | Turbo Variants | | LCM Variants | | ControlNet Variants | | | DR Variants | | | mAcc |
|---|---|---|---|---|---|---|---|---|---|---|---|---|---|---|---|---|---|
| | LDM | SDv1.4 | SDv1.5 | SDv2 | SDXL | SDXL-Refiner | SD-Turbo | SDXL-Turbo | LCM-SDv1.5 | LCM-SDXL | SDv1-Ctrl | SDv2-Ctrl | SDXL-Ctrl | SDv1-DR | SDv2-DR | SDX-L-DR | |
| CNNSpot (Wang et al., 2020) | 99.87 | 99.91 | 99.90 | 97.55 | 66.25 | 86.55 | 86.15 | 72.42 | 98.26 | 61.72 | 97.96 | 85.89 | 82.84 | 60.93 | 51.41 | 50.28 | 81.12 ± 17.6 |
| F3Net (Qian et al., 2020) | 99.85 | 99.78 | 99.79 | 88.66 | 55.85 | 87.37 | 68.29 | 63.66 | 97.39 | 54.98 | 97.98 | 72.39 | 81.99 | 65.42 | 50.39 | 50.27 | 77.13 ± 18.1 |
| CLIP/RN50 (Radford et al., 2021) | 99.00 | 99.99 | 99.96 | 94.61 | 62.08 | 91.43 | 83.57 | 64.40 | 98.97 | 57.43 | 99.74 | 80.69 | 82.03 | 65.83 | 50.67 | 50.47 | 80.05 ± 18.3 |
| GramNet (Liu et al., 2020) | 99.40 | 99.01 | 98.84 | 95.30 | 62.63 | 80.68 | 71.19 | 69.32 | 93.05 | 57.02 | 89.97 | 75.55 | 82.68 | 51.23 | 50.01 | 50.08 | 76.62 ± 17.0 |
| De-fake (Sha et al., 2023) | 92.10 | 99.53 | 99.51 | 89.65 | 64.02 | 69.24 | 92.00 | 93.93 | 99.13 | 70.89 | 58.98 | 62.34 | 66.66 | 50.12 | 50.16 | 50.00 | 75.52 ± 18.4 |
| Conv-B (Liu et al., 2022) | 99.97 | 100.0 | 99.97 | 95.84 | 64.44 | 82.00 | 80.82 | 60.75 | 99.27 | 62.33 | 99.80 | 83.40 | 73.28 | 61.65 | 51.79 | 50.41 | 79.11 ± 18.3 |
| UnivFD (Ojha et al., 2023) | 98.30 | 96.22 | 96.33 | 93.83 | 91.01 | 93.91 | 86.38 | 85.92 | 90.44 | 88.99 | 90.41 | 81.06 | 89.06 | 51.96 | 51.03 | 50.46 | 83.46 ± 17.0 |
| DRCT (Chen et al., 2024a) | 94.45 | 94.35 | 94.24 | 95.05 | 95.61 | 95.38 | 94.81 | 94.48 | 91.66 | 95.54 | 93.86 | 93.48 | 93.54 | 84.34 | 83.20 | 67.61 | 91.35 ± 4.7 |
| Ours/DINOv2 | 99.55 | 99.55 | 99.55 | 99.54 | 99.55 | 94.70 | 99.53 | 99.23 | 99.31 | 99.55 | 99.54 | 99.55 | 99.39 | 99.48 | 99.55 | 97.42 | 99.06 ± 0.1 |
| Ours/CLIP | 99.70 | 99.70 | 99.69 | 99.67 | 99.71 | 99.40 | 99.48 | 99.40 | 99.62 | 99.70 | 99.68 | 99.64 | 99.51 | 99.61 | 99.67 | 97.80 | **99.50** ± 0.1 |

specified, in our proposed Panoptic Patch Learning (PPL), image reconstruction is performed with SDv1.4 inpainting at a generation strength of $s = 0.25$. The inpainting pipeline uses $step = 50$ and guidance scale 7.5. During training, images are randomly cropped to $224 \times 224$, while at test time they are center-cropped to the same resolution. For the randomized patch reconstruction module, the reconstruction patch size is set to $14 \times 14$, consistent with the patch size of ViT. Each fake image in the original training set has a probability of $p_{rpr} = 0.9$ of being replaced with a RPR image, where $r_{rpr} = 50\%$ of patches from a real image are randomly selected to do diffusion reconstruction. For patch-wise contrastive learning, the weight of the contrastive loss is set to $\lambda = 0.3$, with a margin parameter $\alpha = 1.0$.

**Peer methods.** The compared methods involve ResNet-50 (He et al., 2016), Conv-B (Liu et al., 2022), Swin-T (Liu et al., 2021), CNNSpot (Wang et al., 2020), F3Net (Li et al., 2021), SAFE (Li et al., 2024), UnivFD (Ojha et al., 2023), FatFormer (Liu et al., 2024), DRCT (Chen et al., 2024a), C2P-CLIP (Tan et al., 2024a), Effort (Yan et al., 2024b).

## 5.1 COMPARISON WITH OTHER METHODS

**Comparison on GenImage.** Tab. 1 compares PPL with other methods on GenImage. We observe: (1) PPL consistently achieves higher accuracy across different backbones. (2) The standard deviation of PPL's accuracy is smaller, indicating improved stability in detecting diverse generative models.

**Comparison on DRCT-2M.** Tab. 2 reports results on DRCT-2M. The results indicate: (1) PPL consistently achieves SoTA with the lowest std, demonstrating both effectiveness and stability. (2) While DRCT shows relatively poor performance on SDXL-related subsets, PPL maintains a more balanced performance across diverse subsets, underscoring its robustness.

Table 3: Cross-dataset and cross-model accuracy (mAcc) on AIGCDetectionBenchmark. PPL is trained on the GenImage SDv1.4 subset due to its reliance on diffusion-based reconstruction. Baseline methods are trained on ProGAN data provided by AIGCDetectionBenchmark, which is more in-distribution with the test set, thereby giving them an inherent advantage under this setting. Baseline results are taken from AIDE (Yan et al., 2024a).

| Method | ProGAN | StyleGAN | BigGAN | CycleGAN | StarGAN | GauGAN | StyleGAN2 | WFR | ADM | Glide | Midjourney | SD v1.4 | SD v1.5 | VQDM | Wukong | DALLE2 | mAcc |
|---|---|---|---|---|---|---|---|---|---|---|---|---|---|---|---|---|
| CNNSpot | 100.00 | 90.17 | 71.17 | 87.62 | 94.60 | 81.42 | 86.91 | 91.65 | 60.39 | 58.07 | 51.39 | 50.57 | 50.53 | 56.46 | 51.03 | 50.45 | 70.78 ± 18.30 |
| FreDect | 99.36 | 78.02 | 81.97 | 78.77 | 94.62 | 80.57 | 66.19 | 50.75 | 63.42 | 54.13 | 45.87 | 38.79 | 39.21 | 77.80 | 40.30 | 34.70 | 64.03 ± 20.41 |
| Fusing | 100.00 | 85.20 | 77.40 | 87.00 | 97.00 | 77.00 | 83.30 | 66.80 | 49.00 | 57.20 | 52.20 | 51.00 | 51.40 | 55.10 | 51.70 | 52.80 | 68.38 ± 17.46 |
| LNP | 99.67 | 91.75 | 77.75 | 84.10 | 99.92 | 75.39 | 94.64 | 70.85 | 84.73 | 80.52 | 65.55 | 85.55 | 85.67 | 74.46 | 82.06 | 88.75 | 83.84 ± 9.46 |
| LGrad | 99.83 | 91.08 | 85.62 | 86.94 | 99.27 | 78.46 | 85.32 | 55.70 | 67.15 | 66.11 | 65.35 | 63.02 | 63.67 | 72.99 | 59.55 | 65.45 | 75.34 ± 13.8 |
| UnivFD | 99.81 | 84.93 | 95.08 | 98.33 | 95.75 | 99.47 | 74.96 | 86.90 | 66.87 | 62.46 | 56.13 | 63.66 | 63.49 | 85.31 | 70.93 | 50.75 | 78.43 ± 16.19 |
| DIRE-G | 95.19 | 83.03 | 70.12 | 74.19 | 95.47 | 67.79 | 75.31 | 58.05 | 75.78 | 71.75 | 58.01 | 49.74 | 49.83 | 53.68 | 54.46 | 66.48 | 68.68 ± 14.00 |
| DIRE-D | 52.75 | 51.31 | 49.70 | 49.58 | 46.72 | 51.23 | 51.72 | 53.30 | 98.25 | 92.42 | 89.45 | 91.24 | 91.63 | 91.90 | 90.90 | 92.45 | 71.53 ± 20.86 |
| PatchCraft | 100.00 | 92.77 | 95.80 | 70.17 | 99.97 | 71.58 | 89.55 | 85.80 | 82.17 | 83.79 | 90.12 | 95.38 | 95.30 | 88.91 | 91.07 | 96.60 | 89.31 ± 8.61 |
| AIDE | 99.99 | 99.64 | 83.95 | 98.48 | 99.91 | 73.25 | 98.00 | 94.20 | 93.43 | 95.09 | 77.20 | 93.01 | 92.85 | 95.16 | 93.55 | 96.60 | 92.77 ± 7.66 |
| NPR | 99.79 | 97.70 | 84.35 | 96.10 | 99.35 | 82.50 | 98.38 | 65.80 | 69.69 | 78.36 | 77.85 | 78.63 | 78.89 | 78.13 | 76.11 | 64.90 | 82.91 ± 11.54 |
| Ours/DINOv2 | 96.94 | 94.27 | 94.73 | 89.44 | 89.99 | 93.99 | 89.44 | 95.00 | 91.02 | 97.84 | 85.00 | 99.43 | 99.03 | 99.17 | 99.26 | 96.05 | **94.41** ± 4.20 |
| Ours/CLIP | 89.12 | 89.94 | 83.57 | 97.16 | 97.12 | 75.29 | 89.17 | 95.20 | 94.67 | 96.05 | 94.78 | 98.49 | 98.19 | 98.53 | 98.61 | 97.90 | 93.36 ± 6.31 |

Table 4: Cross-dataset and cross-model accuracy (mAcc) on the UniversalFakeDetect. PPL is trained on the SDv1.4 subset of GenImage, while other methods are trained on GAN. Baseline results are taken from C2P-CLIP (Tan et al., 2024a).

| Methods | Ref | GAN | | | | | | Guided | LDM | | | GLIDE | | | DALLE | mAcc |
|---|---|---|---|---|---|---|---|---|---|---|---|---|---|---|---|---|
| | | Pro-GAN | Cycle-GAN | Big-GAN | Style-GAN | Gau-GAN | Star-GAN | | 200 steps | 200 w/cfg | 100 steps | 100 27 | 50 27 | 100 10 | | |
| CNN-Spot | CVPR2020 | 99.99 | 85.20 | 70.20 | 85.70 | 78.95 | 91.70 | 60.07 | 54.03 | 54.96 | 54.14 | 60.78 | 63.80 | 65.66 | 55.58 | 70.05 ± 14.90 |
| Patchfor | ECCV2020 | 75.03 | 68.97 | 68.47 | 79.16 | 64.23 | 63.94 | 67.41 | 76.50 | 76.10 | 75.77 | 74.81 | 73.28 | 68.52 | 67.91 | 71.44 ± 4.73 |
| Co-occurence | Elect. Imag. | 97.70 | 63.15 | 53.75 | 92.50 | 51.10 | 54.70 | 60.50 | 70.70 | 70.55 | 71.00 | 70.25 | 69.60 | 69.90 | 67.55 | 68.78 ± 12.68 |
| Freq-spec | WIFS2019 | 49.90 | 99.90 | 50.50 | 49.90 | 50.30 | 99.70 | 50.90 | 50.40 | 50.40 | 50.30 | 51.70 | 51.40 | 50.40 | 50.00 | 57.55 ± 17.25 |
| F3Net | ECCV2020 | 99.38 | 76.38 | 65.33 | 92.56 | 58.10 | 100.00 | 69.20 | 68.15 | 75.35 | 68.80 | 81.65 | 83.25 | 83.05 | 66.30 | 77.68 ± 12.47 |
| UnivFD | CVPR2023 | 100.00 | 98.50 | 94.50 | 82.00 | 99.50 | 97.00 | 70.03 | 94.19 | 73.76 | 94.36 | 79.07 | 79.85 | 78.14 | 86.78 | 87.69 ± 9.97 |
| LGrad | CVPR2023 | 99.84 | 85.39 | 82.88 | 94.83 | 72.45 | 99.62 | 77.50 | 94.20 | 95.85 | 94.80 | 87.40 | 90.70 | 89.55 | 88.35 | 89.53 ± 7.70 |
| FreqNet | AAAI2024 | 97.90 | 95.84 | 90.45 | 97.55 | 90.24 | 93.41 | 86.70 | 84.55 | 99.58 | 65.56 | 85.69 | 97.40 | 88.15 | 59.06 | 88.01 ± 11.56 |
| NPR | CVPR2024 | 99.84 | 95.00 | 87.55 | 96.23 | 86.57 | 99.75 | 84.55 | 97.65 | 98.00 | 98.20 | 96.25 | 97.15 | 97.35 | 87.15 | 94.37 ± 5.19 |
| FatFormer | CVPR2024 | 99.89 | 99.32 | 99.50 | 97.15 | 99.41 | 99.75 | 76.00 | 98.60 | 94.90 | 98.65 | 94.35 | 94.65 | 94.20 | 98.75 | **96.08** ± 5.95 |
| C2P-CLIP | AAAI2025 | 99.98 | 97.31 | 99.12 | 96.44 | 99.17 | 99.60 | 69.10 | 99.25 | 97.25 | 99.30 | 95.25 | 95.25 | 96.10 | 98.55 | 95.83 ± 7.57 |
| Ours/DINOv2 | | 96.94 | 89.44 | 94.73 | 94.27 | 93.99 | 89.99 | 92.30 | 99.40 | 99.40 | 99.40 | 98.35 | 98.60 | 98.50 | 99.40 | 96.05 ± 3.58 |
| Ours/CLIP | | 89.12 | 97.16 | 83.57 | 89.94 | 75.29 | 97.12 | 94.55 | 98.80 | 98.80 | 98.80 | 96.90 | 97.40 | 97.65 | 98.80 | 93.85 ± 6.78 |

Table 5: Cross-dataset accuracy (Acc) on the in-the-wild dataset Chameleon. Results of other methods are directly taken from AIDE (Yan et al., 2024a). For each training dataset, the first row reports the overall **Acc** on the Chameleon test set, while the second row presents **Acc** separately for **fake images** and **real images** for detailed analysis.

| Training Dataset | CNNSpot | FreDect | Fusing | UnivFD | DIRE | PatchCraft | NPR | AIDE | Ours/CLIP | Ours/DINOv2 |
|---|---|---|---|---|---|---|---|---|---|---|
| SD v1.4 | 60.11 | 56.86 | 57.07 | 55.62 | 59.71 | 56.32 | 58.13 | 62.60 | 63.94 | **66.63** |
| | 8.86/98.63 | 1.37/98.57 | 0.00/99.96 | 17.65/93.50 | 11.86/95.67 | 3.07/96.35 | 2.43/100.00 | 20.33/94.38 | 17.27/99.01 | 64.65/68.12 |
| All GenImage | 60.89 | 57.22 | 57.09 | 60.42 | 57.83 | 55.70 | 57.81 | 65.77 | 69.33 | **72.07** |
| | 9.86/99.25 | 0.89/99.55 | 0.02/99.98 | 85.52/41.56 | 2.09/99.73 | 1.39/96.52 | 1.68/100.00 | 26.80/95.06 | 38.93/92.18 | 49.68/88.99 |

**Comparison on AIGCDetectBenchmark and UniversalFakeDetect.** Tab. 3 and Tab. 4 present results on AIGCDetectBenchmark (Zhong et al., 2024) and UniversalFakeDetect (Ojha et al., 2023), respectively. The results of baseline methods are taken from (Yan et al., 2024a) and (Tan et al., 2024a). We observe that PPL, when trained solely on diffusion-generated data, generalizes effectively to detecting GAN-generated images—even surpassing baseline methods trained directly on GAN data—highlighting PPL's strong generalization capability.

**Comparison on the in-the-wild dataset Chameleon.** Tab. 5 reports results on Chameleon, a challenging dataset comprising diverse images collected from online websites. We observe that most existing methods only marginally exceed the accuracy of random guessing (50%). In contrast, PPL achieves 70% accuracy on Chameleon, demonstrating strong generalization on real-world data.

## 5.2 ROBUSTNESS STUDIES

We conduct robustness experiments on GenImage to evaluate the reliability of our method under common image corruptions. As shown in Fig. 6, both CLIP and DINOv2 backbones sustain high accuracy even under severe JPEG compression and strong Gaussian blur, demonstrating the robustness of our approach.

## 5.3 ABLATION STUDIES

**Ablation on the hyperparameters.** Fig. 7 reports the influence of key hyperparameters in PPL. The results suggest four main observations: (1) PPL achieves peak accuracy at $\lambda = 0.3$. (2) PPL is

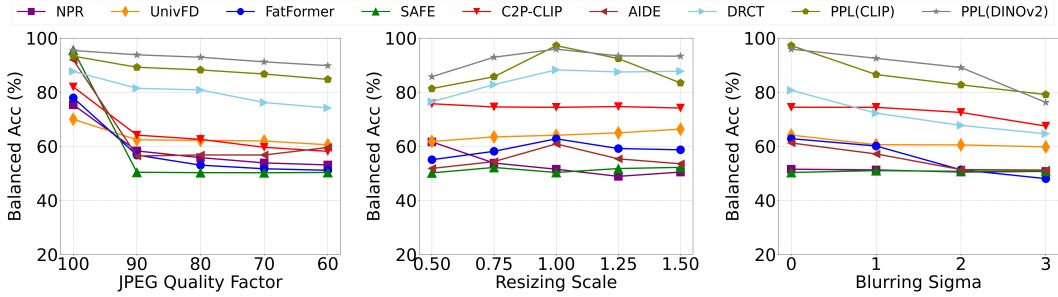

Figure 6: Robustness to image corruptions on GenImage Dataset. Performance is evaluated under JPEG compression (quality factor $Q \in \{100, 90, 80, 70, 60\}$), Gaussian blur (standard deviation $\sigma \in \{0.0, 1.0, 2.0, 3.0\}$), and resizing (scaling factor $S \in \{0.5, 0.75, 1.0, 1.25, 1.5\}$).

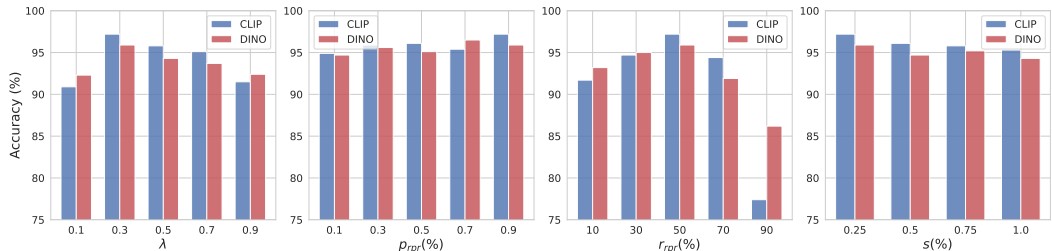

Figure 7: Ablation study on hyperparameters: $\lambda$ representing the weight of contrastive loss. $p_{rpr}$ representing the probability of replacing the fake images with RPR images during training. $r_{rpr}$ representing the ratio of reconstructed patches, and $s$ representing the strength of reconstruction.

relatively robust to $p_{rpr}$, the probability of replacing fake images with RPR images during training. (3) PPL is sensitive to the patch reconstruction ratio $r_{rpr}$, where accuracy degrades at excessively high ratios, with the best performance obtained around $r_{rpr} = 50\%$. (4) Smaller reconstruction strength $s$ not only improves accuracy but also reduces computational cost, making it preferable in practice.

**Ablation on the impact of each module.** Fig. 8 highlights the contributions of Randomized Patch Reconstruction (RPR) and Patch-wise Contrastive Learning (PCL). We compare two strategies for injecting synthetic artifacts into real images: (1) diffusion-based reconstruction (blue), and (2) random patch replacement from original synthetic images (red). The results show that reconstruction serves as a more effective mechanism for introducing synthetic cues to guide the model. While either RPR or PCL alone enhances performance, their combination yields markedly stronger improvements. Additional ablation studies are provided in the Appendix due to space constraints.

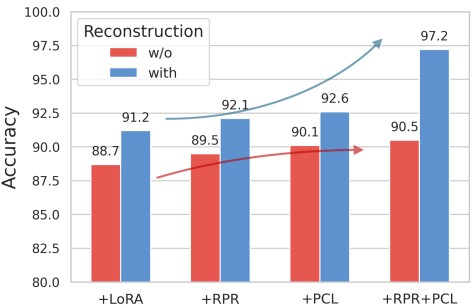

Figure 8: Ablation study on each module.

## 6 CONCLUSION

Our work is based on the nature of the AIGI detection problem, which can be concluded as "All Patches Matter, More Patches Better." However, our observations indicate that existing detectors are unable to fully take advantage of all patches in an AI-generated image. To address this issue, we propose a randomized patch reconstruction augmentation combined with patch-wise contrastive learning strategy. This approach effectively prevents the model from becoming a lazy learner and enhances the utilization of every patch. We achieve state-of-the-art performance on several well-known academic datasets across various benchmark datasets. The outstanding performance achieved in both settings supports our findings and proves the efficacy of the proposed learning framework.

## ACKNOWLEDGMENTS

This work was supported in part by the National Science Foundation for Distinguished Young Scholars under Grant 62225605, "Pioneer" and "Leading Goose" R&D Program of Zhejiang (No.2025C02014), Ningbo Science and Technology Special Projects under Grant No.2025Z028, Project 12326608 supported by NSFC, and the Fundamental Research Funds for the Central Universities.

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

This appendix provides supplementary details and additional analyses. Section A describes the implementation setup; Section B reports more ablation studies; and Section C presents a comparative CDE analysis with both statistical and visual demonstrations.

## A  IMPLEMENTATION DETAILS

**Implementation of RPR.**  In Randomized Patch Reconstruction (RPR), diffusion-reconstructed real images are replaced with their original counterparts on a patch-wise basis. For consistency, we use Stable Diffusion v1 (SDv1) as the reconstruction model and reconstruct real images in the training subset of SDv1.4 from both GenImage and DRCT. The reconstruction is performed with the inpainting pipeline (50 denoising steps, guidance scale 7.5) using an empty prompt, a zero-filled mask of the same size as the input image, and the original image as inputs.

**Implementation of PCL.**  Patch-wise Contrastive Learning (PCL) introduces moderate computational overhead. Training a LoRA model with only binary cross-entropy (BCE) loss on CLIP-Large requires 16 GB of GPU memory, while incorporating the margin-based contrastive loss increases memory usage to 19 GB. Similarly, training one epoch on GenImage with an NVIDIA V100 GPU increases the runtime from 3 to 4 hours.

## B  ADDITIONAL ABLATION STUDIES

We investigate the impact of different contrastive losses applied to embedded patch tokens. Two widely used losses are considered: InfoNCE and the margin-based contrastive loss. InfoNCE maximizes similarity between positive pairs while minimizing it for negative pairs, typically formulated as:

$$\mathcal{L}_{\mathrm{q}} = -\log \frac{\exp(q \cdot k_+ / \tau)}{\sum_{i=0}^{N} \exp(q \cdot k_i / \tau)}, \tag{4}$$

where $q$ and $k_+$ denote the embeddings of the sample and its positive counterpart, and $\tau$ is a temperature parameter. InfoNCE requires computing similarities between each sample and all negative samples, resulting in a computational complexity that grows quadratically with batch size.

In contrast, the margin-based contrastive loss, constrains the Euclidean distance between sample pairs by pulling positive pairs closer and pushing negative pairs apart with a margin $\alpha$:

$$\mathcal{L}_{contrastive} = \sum_{i,j:\ i \neq j} \left[ Y \cdot d_{ij}^2 + (1 - Y) \cdot \max(0, \alpha - d_{ij}^2) \right], \tag{5}$$

where $d_{ij} = \|\mathrm{Emb}_{\mathrm{pat}}^i - \mathrm{Emb}_{\mathrm{pat}}^j\|_2$ is the Euclidean distance between embedded patch tokens, and $Y$ is an indicator function that equals 1 if the pair shares the same label and 0 otherwise. This loss only penalizes negative pairs whose distance is less than the margin, thus avoiding computations over all negative pairs and reducing the overall computational cost. The $d_{ij}$ could also use the cosine distance, which is also compared in our experiments.

**Impact of loss function.**  Tab. 6 illustrates the effectiveness of these loss functions. Unless otherwise specified, all experiments reported in the appendix are conducted on the GenImage dataset, and performance is measured using mean accuracy (mAcc).

**Impact of randomized patch reconstruction vs. fixed position reconstruction.**  Our randomized patch reconstruction method employs a random selection process for image reconstruction, allowing fake patches to appear in different regions across the entire image. Alternatively, replacing patches at fixed positions also yields images composed of both real and synthetic elements. Tab. 7 illustrates the effectiveness of randomized patch reconstruction compared to fixed-position reconstruction, both of which leverage patch-wise contrastive learning.

Table 6: The impact of contrastive loss choice.

| Loss | Midjourney | SDv1.4 | SDv1.5 | ADM | GLIDE | Wukong | VQDM | BigGAN | mAcc |
|---|---|---|---|---|---|---|---|---|---|
| Infonce/$\tau$=0.5 | 93.1 | 99.4 | 99.5 | 89.8 | 96.0 | 99.5 | 99.4 | 89.6 | 95.8 |
| Margin/cosine | 92.8 | 99.7 | 99.5 | 82.3 | 91.5 | 99.7 | 91.5 | 85.6 | 92.8 |
| Margin/euclidean | 94.8 | 98.5 | 98.3 | 94.7 | 96.1 | 98.6 | 98.5 | 98.0 | **97.2** |

Table 7: The impact of random patch replacement vs. fixed position replacement.

| Position | Midjourney | SDv1.4 | SDv1.5 | ADM | GLIDE | Wukong | VQDM | BigGAN | mAcc |
|---|---|---|---|---|---|---|---|---|---|
| Upper Half | 87.1 | 99.7 | 99.6 | 83.9 | 94.5 | 99.7 | 99.2 | 96.0 | 95.0 |
| Lower Half | 87.5 | 99.7 | 99.6 | 81.7 | 93.4 | 99.7 | 99.1 | 94.6 | 94.4 |
| Left Half | 87.7 | 99.8 | 99.5 | 84.4 | 93.8 | 99.7 | 99.0 | 90.6 | 94.3 |
| Right Half | 81.6 | 99.8 | 99.7 | 74.8 | 79.6 | 99.8 | 98.7 | 86.2 | 90.0 |
| Random | 94.8 | 98.5 | 98.3 | 94.7 | 96.1 | 98.6 | 98.5 | 98.0 | **97.2** |

Table 8: The impact of patch size of random patch replacement.

| Patch Size | Midjourney | SDv1.4 | SDv1.5 | ADM | GLIDE | Wukong | VQDM | BigGAN | mAcc |
|---|---|---|---|---|---|---|---|---|---|
| 112 | 99.6 | 99.3 | 92.3 | 92.7 | 99.5 | 95.8 | 99.5 | 94.3 | 96.6 |
| 56 | 99.4 | 99.2 | 93.8 | 90.8 | 99.4 | 95.8 | 99.1 | 97.2 | 96.8 |
| 28 | 98.7 | 98.3 | 95.7 | 95.4 | 98.8 | 97.5 | 98.5 | 98.1 | **97.6** |
| 14 | 94.8 | 98.5 | 98.3 | 94.7 | 96.1 | 98.6 | 98.5 | 98.0 | 97.2 |

Table 9: The impact of random patch replacement vs. random patch dropout.

| Dropout Rate | Midjourney | SDv1.4 | SDv1.5 | ADM | GLIDE | Wukong | VQDM | BigGAN | mAcc |
|---|---|---|---|---|---|---|---|---|---|
| 0.10 | 71.3 | 99.9 | 99.8 | 67.9 | 82.7 | 99.8 | 97.3 | 77.0 | 88.5 |
| 0.15 | 94.3 | 98.7 | 98.6 | 87.9 | 90.4 | 98.7 | 98.5 | 90.9 | 94.2 |
| 0.20 | 77.0 | 99.9 | 99.8 | 69.2 | 72.0 | 99.8 | 98.7 | 87.7 | 89.1 |
| 0.25 | 77.2 | 99.9 | 99.8 | 71.1 | 74.7 | 99.8 | 98.6 | 89.3 | 90.2 |
| Replacement | 94.8 | 98.5 | 98.3 | 94.7 | 96.1 | 98.6 | 98.5 | 98.0 | **97.2** |

**Impact of patch size of randomized patch reconstruction.** Our randomized patch reconstruction method reconstructs real images into fake counterparts where the patch size may influence performance. with patch size potentially influencing performance. Using patches of size $14 \times 14$ is intuitive, as it aligns with the token size used in Vision Transformers (ViTs). Tab. 8 illustrates that the training process is not sensitive to reconstruction patch size, and smaller patch sizes are preferred.

**Impact of randomized patch reconstruction vs. random patch dropout.** Our randomized patch reconstruction method involves substituting fake image patches with their real counterparts, thereby compelling the model to exploit more artifacts from the remaining patches. An alternative approach shown in Fig. 9 is random patch dropout, in which certain patches are removed, resulting in images with fewer patches. Tab. 9 illustrates the effectiveness of Randomized Patch Reconstruction, by comparing patch reconstruction with patch dropout at various dropout rates, with both methods employing patch-wise contrastive learning. The results indicate that, with an appropriate dropout ratio, patch dropout also achieves favorable performance, supporting our hypothesis that models tend to over-rely on a subset of patches. Dropout thus serves as a remedy for this issue. However, patch dropout underperforms patch reconstruction, possibly because patch reconstruction preserves the overall appearance and input domain of the image (i.e., a complete image rather than a masked one), thereby increasing the task's difficulty.

## C  MORE VISUAL COMPARATIVE ANALYSIS

**CDE distribution of different subsets of GenImage.** To better analyze the models' ability to leverage all patches from an image, we use CDE to count the contribution of patch$_{(i,j)}$, which can be

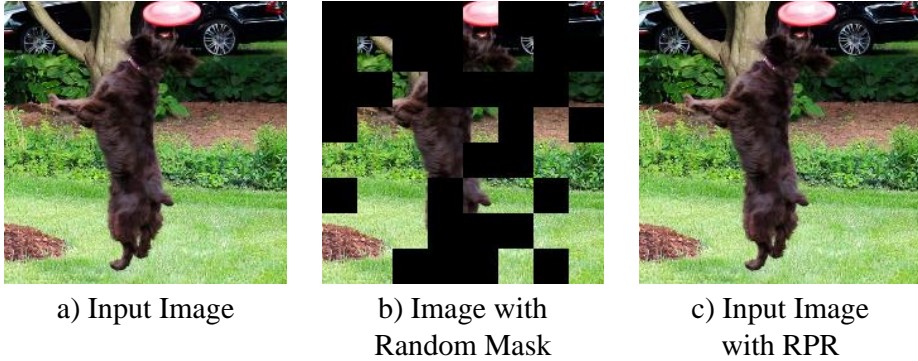

a) Input Image

b) Image with
Random Mask

c) Input Image
with RPR

Figure 9: Visual comparison between random patch dropout (masking) and reconstruction. It is evident that, by reconstruction, the overall visual appearance remains unchanged.

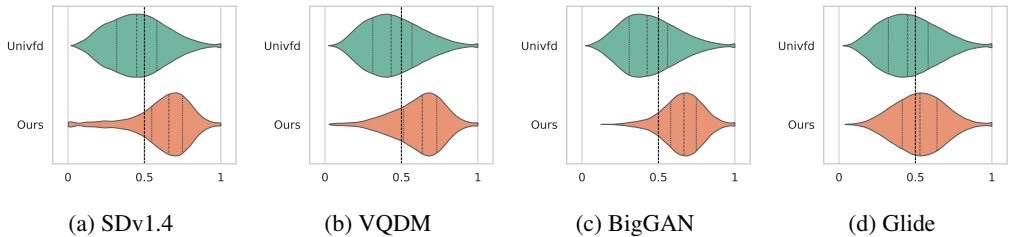

(a) SDv1.4  (b) VQDM  (c) BigGAN  (d) Glide

Figure 10: CDE distribution on different generators of ours and UnivFD.

defined as the difference in logits at the $(i, j)$ position of an image before and after being masked. Fig. 10 illustrates the CDE distribution of UnivFD and our method. For better statistical analysis, we normalize the CDE values to a range $[0, 1]$ using the exponential function $e^{TDE_{(i,j)} - TDE_{max}}$. This normalization facilitates the measurement of differences between less dominant patches and the most dominant patches in the images. The figure demonstrates that a greater number of patches from our method are more uniform.

**Visual showcase of CDE distribution of different subsets on GenImage.**    To better showcase our model's better ability to leverage all patches from an image, we present a visual analysis of CDE across various subsets of the GenImage dataset. The GenImage dataset is divided into multiple subsets, each representing distinct image generation methods. These subsets include GAN-based models such as BigGAN, and diffusion-based models, including Stable Diffusion, VQDM, and ADM. Due to space limitations in the main text, we showcased limited images; here, we present most subset models of GenImage: the diffusion-based Stable Diffusion v1.4 (Fig. 11), the closed-source Midjourney (Fig. 12), and the GAN-based BigGAN (Fig. 13). The rest diffusion-based model are from Fig. 14 to Fig. 16.We use CLIP as backbone for our visualization.

**Visual showcase of GradCAM on GenImage.**    To provide a more intuitive understanding of how our method alters the model's decision-making process, we employ Grad-CAM to visualize the attention maps of the baseline (naive LoRA-tuning) versus our proposed PPL using a CLIP backbone. As illustrated in Fig. 17, the baseline model tends to overfit to sparse, highly localized regions while ignoring the rest of the image. This confirms the prevalence of "Few-Patch Bias" in standard training. In contrast, our method demonstrates a significantly broader spatial coverage, attending to a diverse range of forgery traces across the entire image.

To further investigate the universality of these biases, we extend our visualization to CNN-based architectures. We analyze the activation maps of a ConvNeXt-Base detector with a specific focus on the evolution within the deepest blocks (Stage 3, Block 0,1,2) which are most closely related to decision-making according to (Zeiler & Fergus, 2014). Fig. 18 shows that'Few-Patch Bias' is not

exclusive to ViTs but is a fundamental characteristic of the decision layers in modern CNNs, further justifying the necessity of our proposed learning strategy.

# D  THE USE OF LARGE LANGUAGE MODELS (LLMS)

In this paper, we only use the large language model to help polish our text. The large language model has no role in the research conception.

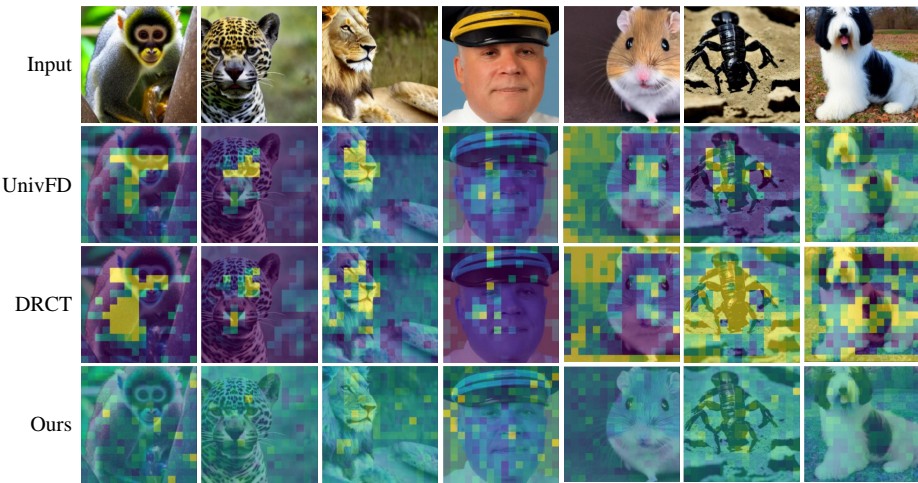

Figure 11: Showcase of CDE map on SDv1.4. Images are sourced from GenImage (Zhu et al., 2024).

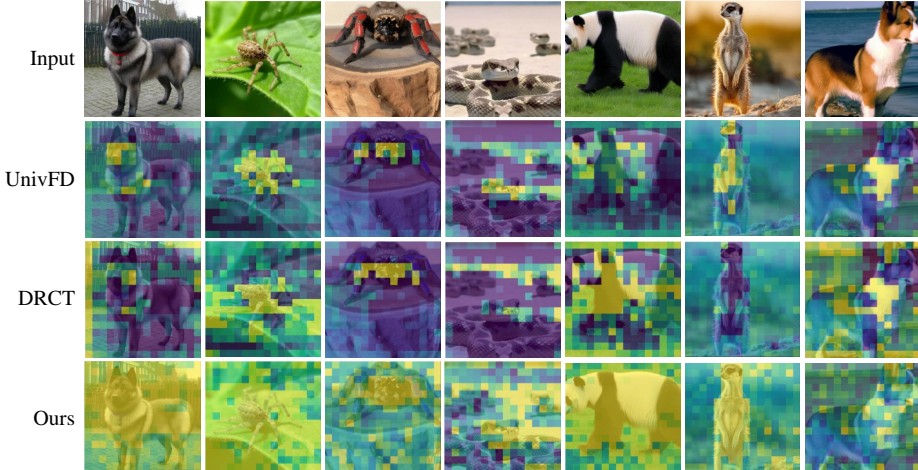

Figure 12: Showcase of CDE map on Midjourney. Images are sourced from GenImage (Zhu et al., 2024).

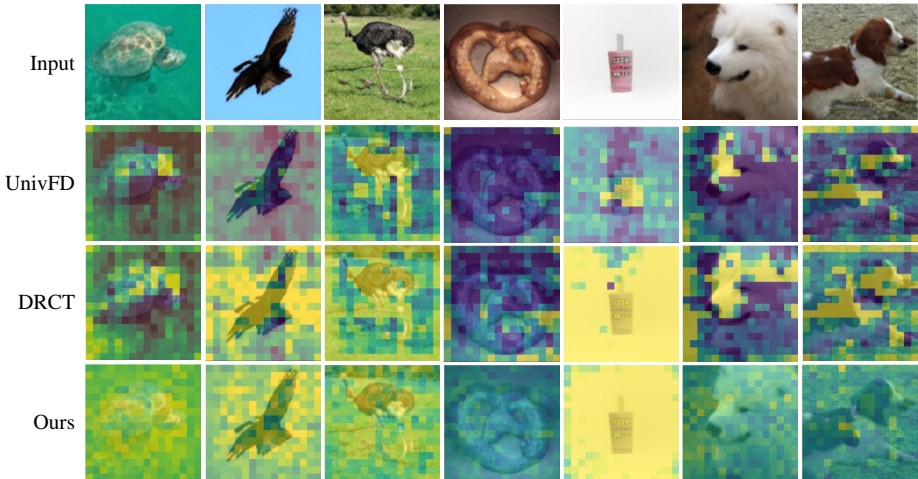

Figure 13: Showcase of CDE map on BigGAN. Images are sourced from GenImage (Zhu et al., 2024).

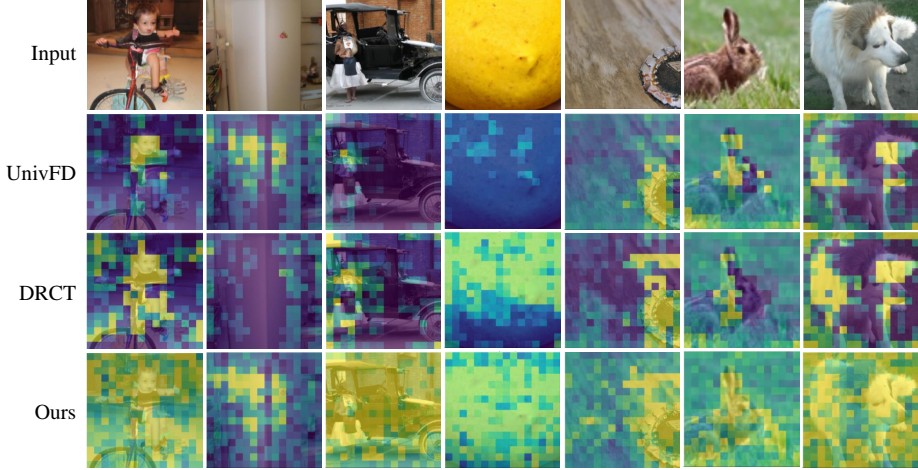

Figure 14: Showcase of CDE map on ADM. Images are sourced from GenImage (Zhu et al., 2024).

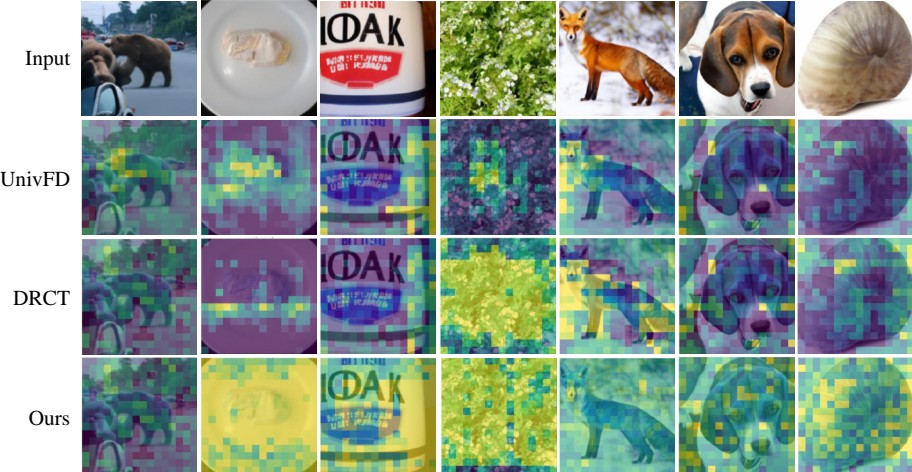

Figure 15: Showcase of CDE map on Glide. Images are sourced from GenImage (Zhu et al., 2024).

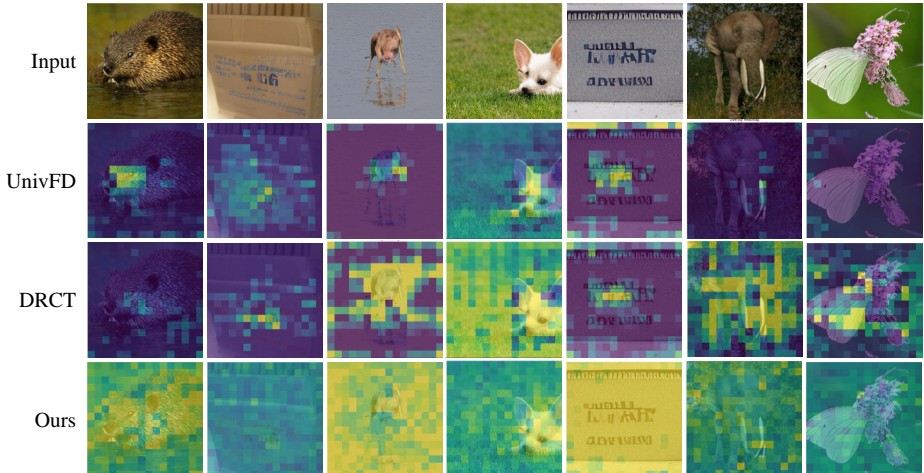

Figure 16: Showcase of CDE map on VQDM. Images are sourced from GenImage (Zhu et al., 2024).

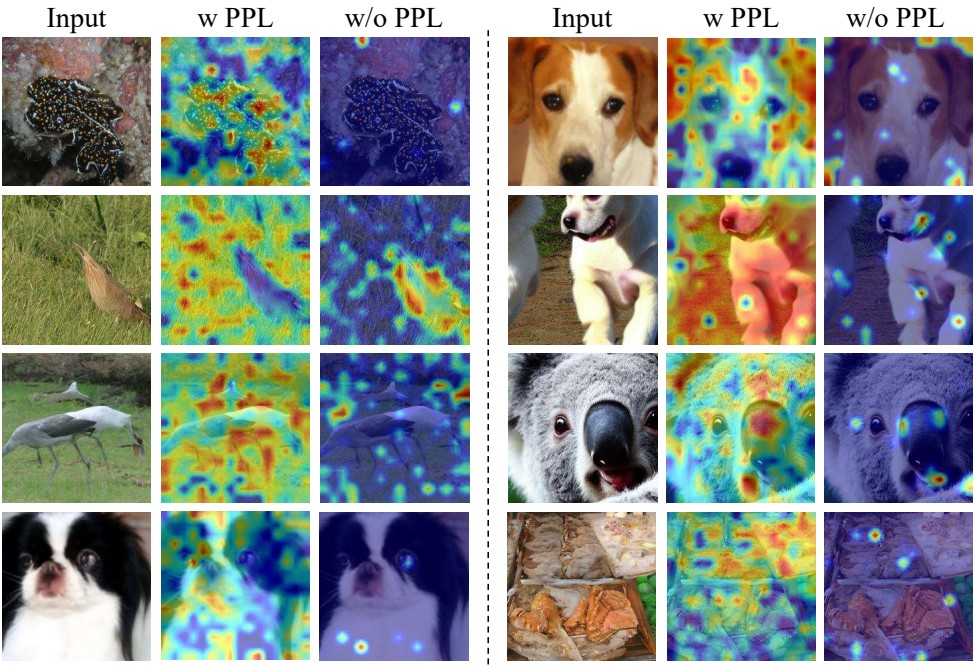

Figure 17: Showcase of GradCAM map of PPL vs naive LoRA-tuning on GenImage. Images are sourced from GenImage (Zhu et al., 2024).

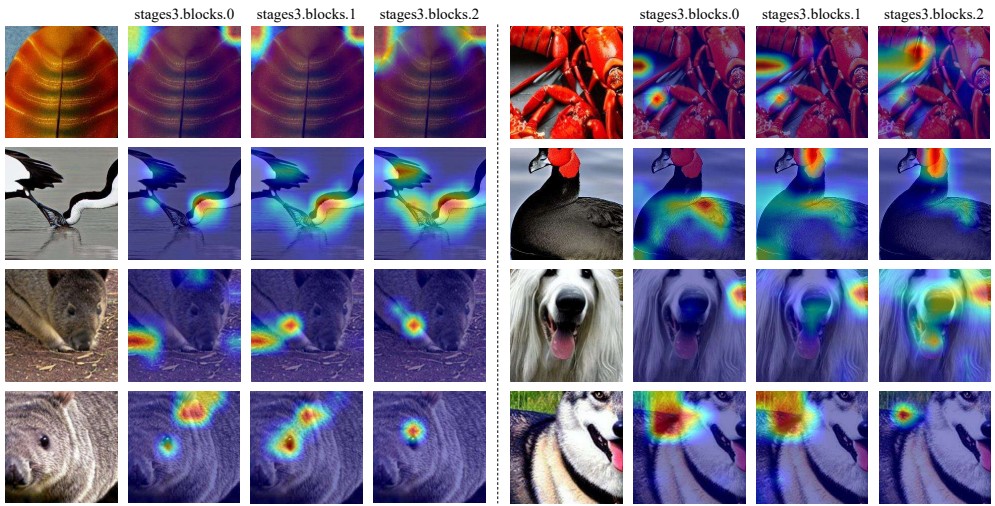

Figure 18: Showcase of GradCAM map on last 3 blocks of ConvNeXt. Images are a sourced from GenImage (Zhu et al., 2024).

