# OpenReview forum: "All Patches Matter, More Patches Better: Enhance AI-Generated Image Detection via Panoptic Patch Learning"
_ICLR.cc/2026/Conference — ICLR 2026 Poster_

### Official Review · Reviewer_JM5D · 2025-10-27

**Soundness:** 3
**Presentation:** 3
**Contribution:** 3
**Rating:** 6
**Confidence:** 4

**Summary:**

This paper identifies a Few-Patch Bias in current AI-generated image detectors and proposes a Panoptic Patch Learning (PPL) framework consisting of (1) Randomized Patch Reconstruction (RPR), reconstructing random real-image patches using Stable Diffusion v1.4 inpainting,  and (2) Patch-wise Contrastive Learning (PCL) applied to ViT patch tokens. The method aims to enforce equal attention to all patches and enhance artifact distribution information. Extensive experiments across several benchmarks show substantial gains.

**Strengths:**

1. The Few-Patch Bias is demonstrated visually (attention/TDE) and via patch-mask counterfactuals. The fix (RPR+PCL) is straightforward and well-motivated.

2. The combination of RPR and PCL provides measurable improvements over existing works.

3. The experiment is conducted on both toy and in-wild datasets, and demonstrates its SOTA performance.

4. There are rich ablation studies to examine the designs of the proposed method.

**Weaknesses:**

1. RPR reconstructs patches of real images using Stable Diffusion v1.4 inpainting (empty prompt). This introduces a major domain-specific bias. It can train additional variants of RPR using GAN-based reconstruction and observe the performance changes. This can further examine the generality of PPL framework. Also, since some baselines, like SAFE, train on GAN-generated images and test on all, such an experiment can make the comparison more fair.

2. In Table 9, Midjourney and ADM accuracies with dropout = 0.15 (94.3% and 87.9%, mAcc=94.2%) are much higher than at 0.1, 0.2, or 0.25 (≈70%), and close to the proposed PPL result (mAcc=97.2%). This raises several questions:

(1) Why does such a narrow dropout range produce a sudden performance surge? Is there randomness in patch masking or seed selection?

(2) If a properly tuned dropout rate (0.15) nearly matches PPL while avoiding diffusion reconstruction and PCL fine-tuning, the claimed advantage of PPL becomes less convincing.

3. The paper fine-tunes CLIP and DINOv2 with PPL but does not clearly report the results of directly training CLIP and DINOv2.

**Questions:**

1. In terms of Weakness-1, (1) have you tried using GAN-based reconstruction or other diffusion models within RPR? (2) And how does the performance change when training RPR with different reconstruction backbones?

2. In terms of Weakness-2, please see the questions in the comment.

3. Could you report the results of directly training or evaluating CLIP/DINOv2 on the same datasets to quantify the actual gain from PPL fine-tuning?

---

> ### Author Response · Authors · 2025-11-27
> **Response to Reviewer JM5D (Part 1)**
>
> We sincerely thank Reviewer JM5D for the thoughtful and constructive feedback. The response is as follows.
>
> ---
>
> > *Q1: RPR reconstructs patches of real images using Stable Diffusion v1.4 inpainting (empty prompt). This introduces a major domain-specific bias. It can train additional variants of RPR using GAN-based reconstruction and observe the performance changes. This can further examine the generality of PPL framework. Also, since some baselines, like SAFE, train on GAN-generated images and test on all, such an experiment can make the comparison more fair.*
>
> **A:** We thank the reviewer for this constructive suggestion regarding experimental fairness and domain bias.
>
> **Fairness of Existing Comparison:** We respectfully clarify that our original experimental setup adheres to standard protocols ensuring fairness:
> * On GenImage and DRCT, both PPL and all baseline methods are trained exclusively on the SDv1.4 subsets.
> * On Chameleon, PPL is trained on the full GenImage dataset like other baselines.
>
> **Generality Validation (Training on GANs):** To rigorously address the concern about domain-specific bias from Stable Diffusion reconstruction, we conducted a new experiment where the entire PPL framework was adapted to a GAN-based pipeline. Specifically, we trained on ProGAN (4-class subset: car, cat, chair, horse) and replaced the reconstruction module with VQGAN.
>
> As shown in Table 1 and Table 2, our method (PPL-VQGAN) consistently outperforms state-of-the-art baselines on both UniversalFakeDetect ($\uparrow$ 0.61%) and AIGCDetect ($\uparrow$ 2.45%), even when trained solely on GAN data. This confirms that the PPL framework is agnostic to the specific generative architecture used for reconstruction.
>
> **Table 1.** $\text{mACC}$ (%) comparison on UniversalFakeDetect. Results sourced from [1].
> |Method|ProGAN|CycleGAN|BigGAN|StyleGAN|GauGAN|StarGAN|Deepfakes|SITD|SAN|CRN|IMLE|Guided|LDM 200|LDM 200 cfg|LDM 100|GLIDE 100 27|Glide 50 27|Glide 100 10|DALLE|mAcc|
> |-|-|-|-|-|-|-|-|-|-|-|-|-|-|-|-|-|-|-|-|-|
> |UniFD|100.0| 98.50| 94.50| 82.00| 99.50| 97.00| 66.60| 63.00| 57.50| 59.5| 72.00| 70.03| 94.19| 73.76| 94.36| 79.07| 79.85| 78.14| 86.78| 81.38|
> |LGrad|97.90| 95.84| 90.45| 97.55| 90.24| 93.41| 97.40| 88.92| 59.04| 71.92| 67.35| 86.70| 84.55| 99.58| 65.56| 85.69| 97.40| 88.15| 59.06| 85.09|
> |FreqNet|99.84| 95.00| 87.55| 96.23| 86.57| 99.75| 76.89| 66.94| 98.63| 50.00| 50.00| 84.55| 97.65| 98.00| 98.20| 96.25| 97.15| 97.35| 87.15| 87.56|
> |FatFormer|99.89| 99.32| 99.50| 97.15| 99.41| 99.75| 93.23| 81.11| 68.04| 69.45| 69.45| 76.00| 98.60| 94.90| 98.65| 94.35| 94.65| 94.20| 98.75| 90.86|
> |C2P-CLIP|99.98| 97.31| 99.12| 96.44| 99.17| 99.60| 93.77| 95.56| 64.38| 93.29| 93.29| 69.10| 99.25| 97.25| 99.30| 95.25| 95.25| 96.10| 98.55| 93.79|
> |PPL-VQGAN|99.91| 98.18| 93.77| 97.64| 94.79| 99.42| 90.88| 75.56| 75.11| 95.6| 99.56| 85.2| 99.7| 99.25| 99.7| 94.75| 94.85| 95.4| 99.45| **94.40**|
>
> **Table 2.** $\text{mACC}$ (%) comparison on AIGCDetect. Results sourced from [2].
> |Method|ProGAN|StyleGAN|BigGAN|CycleGAN|StarGAN|GauGAN|StyleGAN2|WFR|ADM|Glide|Midjourney|SDv1.4|SDv1.5|VQDM|Wukong|DALLE2|mAcc
> |-|-|-|-|-|-|-|-|-|-|-|-|-|-|-|-|-|-|
> | CNNSpot| 100.00 | 90.17| 71.17| 87.62| 94.60| 81.42| 86.91| 91.65 | 60.39 | 58.07 | 51.39| 50.57| 50.53| 56.46 | 51.03| 50.45| 70.78|
> | UnivFD| 99.81| 84.93| 95.08| 98.33| 95.75| 99.47| 74.96| 86.90 | 66.87 | 62.46 | 56.13| 63.66| 63.49| 85.31| 70.93 | 50.75| 78.43|
> | DIRE-G| 95.19| 83.03| 70.12| 74.19| 95.47| 67.79| 75.31| 58.05 | 75.78 | 71.75 | 58.01| 49.74| 49.83| 53.68 | 54.46| 66.48| 68.68 |
> | DIRE-D| 52.75| 51.31| 49.70| 49.58| 46.72| 51.23| 51.72| 53.30 | 98.25 | 92.42 | 89.45| 91.24| 91.63| 91.90 | 90.90| 92.45| 71.53 |
> |NPR| 99.79| 97.70|84.35| 96.10|99.35|82.50| 98.38| 65.80 | 69.69 | 78.36 | 77.85| 78.63| 78.89| 78.13 | 76.11| 64.90|82.91|
> | PatchCraft | 100.00 | 92.77| 95.80| 70.17| 99.97| 71.58| 89.55| 85.80 | 82.17 | 83.79 | 90.12| 95.38| 95.30| 88.91 | 91.07| 96.60| 89.31 |
> |PPL-VQGAN|99.94|99.59|90.08|97.05|99.17|89.19|96.93|95.8|88.98|92.13|64.03|96.13|95.74|95.17|96.14|77.25|**91.76 ($\uparrow$ 2.45)**|

---

> > ### Comment · Reviewer_JM5D · 2025-11-28
> >
> > Thanks for the point-by-point response. My main concerns are about the experiment setting (W1), dropout rate selection and tuning (W2), and CLIP and DINOv2 training (W3). The authors have conducted new experiments and examinations based on my comments and suggestions, and the newly added results demonstrate the effectiveness. As a result, I changed my rating from weak accept to accept.

---

> ### Author Response · Authors · 2025-11-27
> **Response to Reviewer JM5D (Part 2)**
>
> > *Q2: In Table 9, Midjourney and ADM accuracies with dropout = 0.15 (94.3% and 87.9%, mAcc=94.2%) are much higher than at 0.1, 0.2, or 0.25 (≈70%), and close to the proposed PPL result (mAcc=97.2%). This raises several questions:
> (1) Why does such a narrow dropout range produce a sudden performance surge? Is there randomness in patch masking or seed selection?
> (2)If a properly tuned dropout rate (0.15) nearly matches PPL while avoiding diffusion reconstruction and PCL fine-tuning, the claimed advantage of PPL becomes less convincing.*
>
> **A:** We thank the reviewer for this sharp observation regarding the dropout ablation.
>
> Regarding the performance surge: We attribute the performance spike at dropout rate 0.15 to the stochastic nature of random patch masking, which can occasionally align with discriminative features by chance in a single run.
>
> Regarding stability and PPL’s advantage: To address the concern about stability and prove PPL's advantage, we repeated the experiments for each dropout rate ($0.10, 0.15, 0.20, 0.25$) over 5 independent runs with different random seeds. As shown in Table 3, the "Dropout 0.15" baseline exhibits high standard deviation, confirming its instability. In contrast, PPL achieves the highest mean accuracy (96.9%) with significantly lower standard deviation ($\pm 0.64$), demonstrating that our method provides a robust, systematic improvement superior to hyperparameter-tuned dropout.
>
> **Table 3.** Ablation test with different dropout rates.
> |Dropout Rate|SDv1.4|SDv1.5|Midjourney|ADM|Wukong|GLIDE|VQDM|BigGAN|mACC|
> |-|-|-|-|-|-|-|-|-|-|
> |0.10|99.6±0.08|99.6±0.07|76.8±5.15|68.4±8.71|99.7±0.04|80.7±3.42|95.5±2.00|85.4±5.82|88.2±2.30|
> |0.15|99.8±0.10|99.7±0.08|84.6±5.43|72.3±9.27|99.7±0.07|84.0±3.92|97.8±1.98|89.3±6.66|90.9±3.28|
> |0.20|99.6±0.11|99.6±0.10|76.3±3.39|66.5±5.93|99.8±0.10|76.8±4.90|94.1±1.49|86.5±2.67|87.4±2.09|
> |0.25|99.8±0.06|99.7±0.08|76.5±5.59|67.5±3.70|99.7±0.07|75.9±4.29|86.6±3.81|82.7±3.05|86.0±2.15|
> |PPL|98.2±0.09|97.7±0.09|94.8±2.80|94.5±1.80|98.0±0.08|97.1±0.45|97.9±0.25|97.0±0.50|**96.9±0.64**|
>
> ---
>
> > *Q3: The paper fine-tunes CLIP and DINOv2 with PPL but does not clearly report the results of directly training CLIP and DINOv2.*
>
> **A:** We appreciate the reviewer for pointing out this omission.
>
> We have now included the results for naively fine-tuning CLIP and DINOv2 (using LoRA) on the GenImage dataset. To ensure a fair comparison, these baselines were trained using the same reconstruction data utilized in our pipeline.
>
> The results indicate that naive fine-tuning yields significantly lower performance compared to our approach. This substantial performance gap validates the effectiveness of the PPL framework in learning more robust discriminative features.
>
> **Table 4.** $\text{mACC}$ (%) comparison with naive trained baseline on GenImage
> | Method | SDv4 | SDv5 | Midjourney | ADM | Wukong | Glide | VQDM | BigGAN | Average |
> |-|-|-|-|-|-|-|-|-|-|
> | Dino-LoRA Naive |98.2|97.9|72.3|79.4| 97.8| 87.2| 92.9| 88.9| 89.3|
> |DINO-PPL|98.2|97.7|90.4|91.8|98.0|96.3|97.7|96.2|**95.9**|
> | CLIP-LoRA Naive | 99.8 | 99.7 | 87.4 | 69.8 | 99.7 | 88.2 | 97.5 | 86.9 | 91.1 |
> |CLIP-PPL|98.5|98.3|94.8|94.7|98.6|96.1|98.5|98.0|**97.2**|
>
> ---
>
> [1] C2P-CLIP: Injecting Category Common Prompt in CLIP to Enhance Generalization in Deepfake Detection, AAAI 2025.
>
> [2] PatchCraft: Exploring Texture Patch for Efficient AI-generated Image Detection. Arxiv.

---

### Official Review · Reviewer_sAHA · 2025-10-27

**Soundness:** 2
**Presentation:** 3
**Contribution:** 3
**Rating:** 6
**Confidence:** 5

**Summary:**

This paper proposed a novel patch-based AI-generated image detection method. The main motivation is the few-patch bias observed in existing detectors, i.e., they overly rely on a limited proportion of patches and neglect the diversity of artifacts across patches. To encourage the utilization of information from all patches, Randomized Patch Reconstruction (RPR) is proposed, which applies diffusion reconstruction to real images and replaces a random set of the original image patches with the corresponding reconstructed patches. Patch-wise Contrastive Learning (PCL) further encourages the learning and utilization of all patch features. Results on several benchmarks suggest the state-of-the-art generalization performance of the proposed method.

**Strengths:**

1. The analysis of the few-patch bias of AIGI detectors reveals a significant limitation of existing methods.
2. The proposed RPR and PCL effectively encourage the model to utilize the information across all patches.
3. The generalization of the proposed method is comprehensively evaluated, including testing on challenging datasets like Chameleon and robustness studies.

**Weaknesses:**

1. The motivation of the *All Patches Matter* principle requires further clarification.
- The two lines of evidence stated in Lines 45-49 only support that "some patches matter" (i.e., some of the patches contain discriminative patterns) rather than "all patches matter".
- The "Theory" in Line 146 may be an inappropriate title for the first key finding, as no theoretical results are provided. In addition, the assumptions behind the statement "Because every patch of a synthetic image is itself generated, each inherently contains artifacts" need further clarification (e.g., what "artifacts" are and why generative models produce them across every pixel).
- The details for the "Experiments" in Lines 154-157 are not specified.
- Given that "a single patch contains sufficient information for reliable discrimination" (Line 157), it seems unnecessary to emphasize the utilization of all patches. This point may not support the *All Patches Matter* principle.
2. It seems that the attention maps in Figure 3(a) can be explained by the observations in [1] that vision transformers tend to utilize patch tokens in low-informative background areas as registers for aggregating global information. Repeating the visualization experiments with vision transformers with dedicated registers proposed in [1] may eliminate this possibility. Besides, the acquisition of the attention maps needs explanation.
4. The experimental details for Figure 3(b) are not specified, including what detectors are tested and how the patch masking is implemented. This is important for supporting the generalization of the conclusion.
5. It seems that the Total Direct Effect (TDE) described in Lines 210-213 should be the Controlled Direct Effect (CDE).
6. Previous reconstruction-based methods such as [2] and [3] should be discussed and compared.


[1] Vision Transformers Need Registers. ICLR 2024.
[2] Aligned Datasets Improve Detection of Latent Diffusion-Generated Images. ICLR 2025.
[3] A Bias-Free Training Paradigm for More General AI-generated Image Detection. CVPR 2025.

**Questions:**

1. How are the reconstructed images (in Figure 2 and Section 4) produced? Is there any image processing process, such as upsampling and downsampling, that can affect the low-level details of the image or introduce artifacts?
2. Why does the proposed method generalize effectively to GAN-generated images, despite the training is solely based on diffusion models?
3. Is it possible to set $p_{rpr}$ to 1, i.e., using only the real images and the RPR images for training?
4. In Figure 8, what contributes to the difference between the blue and red bars at +LoRA, given that RPR is not used?

---

> ### Author Response · Authors · 2025-11-27
> **Response to Reviewer sAHA (Part 1)**
>
> We sincerely thank Reviewer sAHA for the thoughtful and constructive feedback. The response is as follows.
>
> ---
>
> > *Q1: The two lines of evidence stated in Lines 45-49 only support that "some patches matter" (i.e., some of the patches contain discriminative patterns) rather than "all patches matter".*
>
> **A:** We appreciate this insightful comment.
>
> As clarified in Main paper, Footnote 1, "This work adheres to the mainstream AIGI detection setting, where the entire image is generated by AI models". Consequently, every patch contains generative artifacts and contributes discriminative information. We clarify that "All Patches Matter" refers to the availability of forensic evidence in every patch, whereas "More Patches Better" refers to the necessity of utilizing them to prevent detectors from overfitting to specific, easy-to-learn local features
>
> ---
>
> > *Q2: The details for the "Experiments" in Lines 154-157 are not specified.*
>
> **A:** We thank the reviewers for questioning the lack of details of the experiment related with the patch experiments.
>
> To validate the prevalence of artifacts, we randomly cropped a $28 \times 28$ patch from each image before feeding it to the network. The detector utilized was an $\text{RN50}$ trained on the SDv1.4 subset of the GenImage dataset. Table 1 confirms that even a single arbitrary patch contains reliable information for discrimination.
>
> **Table 1.** $\text{mACC}$ (%) performance of a single patch on GenImage.
> |SDv1.4|SDv1.5|Midjourney|ADM|Wukong|GLIDE|VQDM|BigGAN|mACC|
> |-|-|-|-|-|-|-|-|-|
> |99.2|99.2|88.5|91.0|99.2|85.3|96.8|54.3|89.6|
>
> ---
>
> > *Q3: Given that "a single patch contains sufficient information for reliable discrimination" (Line 157), it seems unnecessary to emphasize the utilization of all patches. This point may not support the All Patches Matter principle.*
>
> **A:** We appreciate the reviewer’s careful observation. We clarify that while a single patch is sufficient for detection in simple scenarios, utilizing more patches is optimal for robust generalization.
>
> Following the experiment detailed in Q2, we conducted a further comparative study: We randomly sampled and combined 16 non-overlapping $28 \times 28$ patches to form a squared image, which was then used to train an $\text{RN50}$ detector on the SDv1.4 subset of GenImage.As shown in Table 2, simply increasing patch count yields only a marginal improvement ($\uparrow$ 2.7%). However, our proposed PPL framework, which encourages the utilization of all patches, achieves a significant performance boost (+7.6%), particularly in cross-generator tasks. This validates that explicitly leveraging distributed artifacts is essential for generalizability.
>
> **Table 2.** $\text{mACC}$ (%) performance of a single patch on GenImage.
> |Method|SDv1.4|SDv1.5|Midjourney|ADM|Wukong|GLIDE|VQDM|BigGAN|mACC|
> |-|-|-|-|-|-|-|-|-|-|
> |1 patch|99.2|99.2|88.5|91.0|99.2|85.3|96.8|54.3|89.6|
> |16 patches|97.9|98.1|92.1|91.7|98.1|95.8|94.4|67.9|92.3 ($\uparrow$ 2.7)|
> |PPL(all patch)|98.5|98.3|94.8|94.7|98.6|96.1|98.5|98.0|97.2 **($\uparrow$ 7.6)**|
>
> ---
>
> > *Q4: The "Theory" in Line 146 may be an inappropriate title for the first key finding, as no theoretical results are provided. In addition, the assumptions behind the statement "Because every patch of a synthetic image is itself generated, each inherently contains artifacts" need further clarification (e.g., what "artifacts" are and why generative models produce them across every pixel).*
>
> **A:** We appreciate the reviewer’s feedback and agree that "Theory" was an imprecise descriptor. We have renamed it to "Principle" (line 146, page 3) to better reflect the nature of our findings.
>
> What "artifact" are: As established in prior work [1], the upsampling operations inherent to GANs and Diffusion Models introduce distinct structural patterns, namely, "neighbouring pixel relationships" that differ significantly from the natural images. Following [1], define these unique patterns as generative artifacts.
>
> Why across every patch: Since our work focus on images wholly generated, each part of an image could contain fake artifacts. Our single patch experiment (Table 1) proved that that fake artifacts lies in any possible patch in an image.

---

> > ### Author Response · Authors · 2025-11-27
> > **Response to Reviewer sAHA (Part 2)**
> >
> > > *Q5: It seems that the attention maps in Figure 3(a) can be explained by the observations in [2] that vision transformers tend to utilize patch tokens in low-informative background areas as registers for aggregating global information. Repeating the visualization experiments with vision transformers with dedicated registers proposed in [2] may eliminate this possibility. Besides, the acquisition of the attention maps needs explanation.*
> >
> > **A:** We appreciate the reviewer connecting our observations to the "register tokens" phenomenon described in [2].
> >
> > Attention Map Extraction: To generate Figure 3(a), we visualized the attention weights of the [CLS] token from the final block of the ViT backbone. The procedure is as follows:
> >
> > * **Extraction:** We extract the attention matrix $\mathbf{A} \in \mathbb{R}^{N_h \times N_s \times N_s}$ from the self-attention mechanism of the final layer ($N_h$: heads, $N_s$: sequence length).
> > * **Filtering:** We isolate the attention weights connecting the [CLS] token (index 0) to all subsequent spatial tokens, yielding $\mathbf{A}^{\text{cls}} \in \mathbb{R}^{N_h \times N_{p}}$ ($N_p$: number of patches).
> > * **Aggregation & Projection:** We average the weights across all heads to obtain $\mathbf{A}^{\text{map}} \in \mathbb{R}^{N_{p}}$, which is then reshaped and bilinearly upsampled to the original image resolution.
> >
> > To determine if the observed bias was merely an architectural artifact, we repeated the experiments using the official DINOv2 with registers [2].
> >
> > As shown in Table 3, while registers mitigate sparse attention artifacts, they yield only a marginal performance improvement ($\uparrow$ 1.0%). In contrast, our PPL method achieves a significant boost (95.9%), validating that our approach offers a more direct and effective solution.
> >
> > **Table 3.** $\text{mACC}$ (%) comparison of DINOv2 with registers on GenImage.
> > |Method|SDv1.4|SDv1.5|Midjourney|ADM|Wukong|Glide|VQDM|BigGAN|mACC|
> > |-|-|-|-|-|-|-|-|-|-|
> > |DINOv2|99.3|99.1|81.4|58.3|97.6|76.9|77.4|60.1|81.3|
> > |DINOv2+register|99.8|99.7|81.3|57.9|99.0|77.6|85.0|58.3|82.3 ($\uparrow$ 1)|
> > |DINOv2+PPL|99.4|99.0|85.0|91.0|99.3|97.8|99.2|96.5|95.9|
> > |DINOv2+register+PPL|97.1|96.8|91.8|92.5|97.0|96.5|96.9|96.5|95.6|
> >
> > ---
> >
> > > *Q6: The experimental details for Figure 3(b) are not specified, including what detectors are tested and how the patch masking is implemented. This is important for supporting the generalization of the conclusion.*
> >
> > **A:** We thank the reviewer for highlighting this omission. We have updated the manuscript with the following details:
> > * **Detector:** The analysis was conducted using UnivFD (CLIP backbone) trained on the GenImage dataset.
> > * **Masking Implementation:** We employed zero-masking, where pixel values within the target patch are set to zero.
> > * **Metric Calculation:** For each image, we iteratively masked every patch (sizes ranging from $14 \times 14$ to $112 \times 112$) and recorded the drop in the target-class (fake) logit. In Figure 3(b), the "Impact Range" visualizes the maximum logit drop (most sensitive patch) and the minimum logit drop (least sensitive patch), while the middle curve represents the average drop across all patches.
> > * **Evaluation Metric:** The reported performance metric is the Recall Rate on the GenImage test set.
> >
> > ---
> >
> > > *Q7: It seems that the Total Direct Effect (TDE) described in Lines 210-213 should be the Controlled Direct Effect (CDE).*
> >
> > **A:** We are sincerely grateful to the reviewer for this precise correction regarding causal inference terminology. We have revisited the formal definitions:
> >
> > $$CDE(m) = E[Y(a, m) - Y(a^*, m)]$$
> >
> > In our experimental setting (Equation 1 in official paper):
> > * $a$ (Treatment): The target patch $(i,j)$ is present.
> > * $a^*$ (Control): The target patch $(i,j)$ is masked.
> > * $m$ (Context/Mediators): All other patches in the image, which are held fixed at their original pixel values.
> >
> > Our metric measures the change in logits by masking the target patch $a$ while explicitly fixing the context $m$. This strictly aligns with the definition of Controlled Direct Effect (CDE)[3].
> >
> > Accordingly, we have corrected the terminology from TDE to CDE in the revised manuscript. This change is terminological and does not alter the computational methodology or experimental conclusions.

---

> ### Author Response · Authors · 2025-11-27
> **Response to Reviewer sAHA (Part 3)**
>
> > *Q8: Previous reconstruction-based methods such as [4] and [5] should be discussed and compared.*
>
> **A:**  We appreciate the reviewer for bringing these relevant reconstruction-based works to our attention.
>
> While methods like [4] and [5] typically rely on whole-image reconstruction, our method employs a randomized local patch injection strategy. This mechanism explicitly encourages the model to capture fine-grained artifact patterns that global methods may overlook.
>
> As demonstrated in our comparative analysis, our method achieves significant performance gains over these baselines, with improvements of **14.2%** (Table 3) and **6.0%** (Table 4). These results validate the superior generalizability of our patch-based approach.
>
> **Table 4.** $\text{mACC}$ (%) comparison with Aligned on GenImage. Results of Aligned are based on our re-implementation with official
> | Method | SDv1.4 | SDv1.5 | Midjourney | ADM | Wukong | GLIDE | VQDM | BigGAN | mACC |
> | - | - | - | - | - | - | - | - | - | - |
> | Aligned | 99.9 | 99.8 | 93.8 | 50.9 | 99.9 | 54.1 | 70.6 | 50.7 | 77.5 |
> | Aligned-Shaders | 93.3 | 93.0 | 90.0 | 58.0 | 93.0 | 59.4 | 67.7 | 61.3 | 77.0 |
> | Aligned-Sync | 99.8 | 99.7 | 96.3 | 54.0 | 99.7 | 63.3 | 74.8 | 51.8 | 79.9 |
> | Aligned-Sync-Shaders | 92.0 | 91.7 | 88.8 | 61.7 | 91.3 | 60.3 | 70.3 | 65.3 | 77.7 |
> | Aligned trained on Genimage | 99.8 | 99.8 | 96.5 | 60.5 | 99.8 | 62.0 | 82.0 | 63.6 | 83.0 |
> |ours/CLIP|98.5|98.3|94.8|94.7|98.6|96.1|98.5|98.0|**97.2 ($\uparrow$ 14.2)**|
>
>
> **Table 5.** $\text{mACC}$ (%) comparison with Bias-Free on unbiased GenImage. Results of Bias-Free are sourced from [5].
> | Method | SDv1.4 | SDv1.5 | Midjourney | ADM | Wukong | GLIDE | VQDM | BigGAN | mACC |
> | - | - | - | - | - | - | - | - | - | - |
> |Bias-Free|98.8|98.8|95.7|79.8|99.0|85.3|88.7|68.7|89.3|
> |ours/CLIP|98.0|97.7|88.8|83.1|97.9|91.1|95.3|92.8|**93.1 ($\uparrow$ 3.8)**|
> |ours/DINO|98.2|97.7|88.7|91.5|98.0|95.2|97.5|95.7|**95.3 ($\uparrow$ 6.0)**|
>
> ---
>
> > *Q9: How are the reconstructed images (in Figure 2 and Section 4) produced? Is there any image processing process, such as upsampling and downsampling, that can affect the low-level details of the image or introduce artifacts?*
>
> **A:** We appreciate the opportunity to clarify the details of our data generation pipeline.
>
> As detailed in Section A of Appendix (Line 655-659), we utilize the SDv1.4 as inpainting model. The generation parameters are set as follows: an empty prompt, a zero-filled mask, a denoising strength of **$s=0.25$**, **50** inference steps, and a guidance scale of **7.5**.
>
> Regarding the potential introduction of artifacts, we explicitly avoid resizing operations which could distort low-level forensic details. Instead, we pad input images before reconstruction to ensure their dimensions are multiples of 8, and subsequently crop them to their original resolution after reconstruction. This approach strictly preserves the integrity of the original high-frequency artifacts without introducing upsampling or downsampling noise.
>
> ---
>
> > *Q10: Why does the proposed method generalize effectively to GAN-generated images, despite the training is solely based on diffusion models?*
>
> **A:** We appreciate this question exploring the mechanism of our cross-architecture generalization.
>
> This capability is attributed to the structural similarities in the generation process. As established in [1], the up-sampling operations utilized by both GANs and Diffusion Models introduce shared, generalized spectral artifacts. Consequently, by training on these localized artifacts in diffusion models, our method learns robust features that transfer effectively to GAN-based generators.
>
> ---
>
> > *Q11: Is it possible to set  to 1, i.e., using only the real images and the RPR images for training?*
>
> **A:** We thank the reviewer for this suggestion to verify the impact of synthetic data sources.
>
> We conducted an experiment where the model was trained exclusively on real images (from the GenImage SDv1.4 subset) and their corresponding RPR (diffusion-reconstructed) counterparts, removing the original "fake" images entirely.
>
> As shown in Table 6, this configuration yields results comparable to the full training set. This confirms that our reconstruction-based artifacts alone are sufficient to guide the model in learning robust discriminative features.
>
> **Table 6.** $\text{mACC}$ (%) comparison with and without original fake images on GenImage.
> | Method | SDv4 | SDv5 | Midjourney | ADM | Wukong | Glide | VQDM | BigGAN | Average |
> | - | - | - | - | - | - | - | - | - | - |
> | CLIP-PPL all rec | 99.5 | 99.5 | 86.0 | 83.1 | 99.6 | 91.8 | 99.4 | 94.7 | 94.2|
> |CLIP-PPL|98.5|98.3|94.8|94.7|98.6|96.1|98.5|98.0|97.2|
> | Dino-PPL all rec | 98.9 | 98.5 | 70.2 | 87.8 | 98.6 | 87.3 | 97.7 | 94.7 | 91.7|
> |DINO-PPL|98.2|97.7|90.4|91.8|98.0|96.3|97.7|96.2|95.9|

---

> ### Author Response · Authors · 2025-11-27
> **Response to Reviewer sAHA (Part 4)**
>
> > *Q12: In Figure 8, what contributes to the difference between the blue and red bars at +LoRA, given that RPR is not used?*
>
> **A:** We thank the reviewer for observing this detail.
>
> The distinction between the blue and red bars lies in the data augmentation strategy employed throughout the ablation study.
>
> As described in Section 5.3, the blue bars utilize our proposed Diffusion Reconstruction strategy, while the red bars utilize the baseline Random Patch Replacement with patches from original synthetic images.
>
> Therefore, at the +LoRA baseline step, the blue bar represents LoRA-tuning with additional diffusion reconstructed images, whereas the red bar represents LoRA-tuning with original synthetic images.
>
> ---
>
> [1] Rethinking the Up-Sampling Operations in CNN-based Generative Network for Generalizable Deepfake Detection. CVPR 2024.
>
> [2] Vision Transformers Need Registers. ICLR 2024.
>
> [3] Direct and indirect effects. UAI 2001.
>
> [4] Aligned Datasets Improve Detection of Latent Diffusion-Generated Images. ICLR 2025.
>
> [5] A Bias-Free Training Paradigm for More General AI-generated Image Detection. CVPR 2025.

---

### Official Review · Reviewer_EuiK · 2025-10-29

**Soundness:** 3
**Presentation:** 3
**Contribution:** 3
**Rating:** 8
**Confidence:** 4

**Summary:**

This paper tackles the challenge of generalizing AI-generated image (AIGI) detectors across different generation models. Through systematic analysis, the authors identify a key issue — “Few-Patch Bias” — where existing detectors over-rely on a small number of image patches despite artifacts being uniformly distributed across all regions of synthetic images. They propose two guiding principles, *All Patches Matter* and *More Patches Better*, and introduce the **Panoptic Patch Learning (PPL)** framework to operationalize them. PPL combines **Randomized Patch Reconstruction (RPR)**, which injects synthetic artifacts into randomly chosen patches to diversify learning, and **Patch-wise Contrastive Learning (PCL)**, which enforces consistent discriminative capability across patches. Extensive experiments on major benchmarks (GenImage, DRCT-2M, AIGCDetectionBenchmark, and Chameleon) demonstrate that PPL achieves state-of-the-art accuracy and robustness, significantly improving generalization to unseen generators and real-world data.

**Strengths:**

1. The paper introduces clear and insightful principles (“All Patches Matter” and “More Patches Better”) that reveal a fundamental property of AI-generated images and motivate the proposed framework.
2. The proposed Panoptic Patch Learning method is conceptually simple yet effective, combining randomized patch reconstruction and patch-wise contrastive learning to mitigate few-patch bias.
3. Extensive experiments across diverse benchmarks demonstrate strong generalization and robustness, supported by thorough analyses and clear visual evidence.

**Weaknesses:**

1. In the second paragraph of the introduction, the term *“patch”* appears for the first time but lacks a clear definition or motivation. Since AIGI detection includes many CNN-based detectors that do not explicitly rely on patch-level representations, introducing the patch concept without clarification may confuse readers about its relevance to this task. It is recommended that the authors explain why they adopt patch as the basic analytical unit and cite related works that have previously used patch-based approaches, which would make the motivation more convincing.
2. In line 364, the reference of SAFE is wrong.
3. In Tables 3 and 4, I notice that all baseline methods are trained on GAN-based datasets, while the proposed PPL is consistently trained on SDv1.4 (a diffusion-based model). Although it is reasonable to fix one generator for evaluating generalization, different training datasets may have biases toward different test distributions (e.g., a model trained on diffusion data may generalize better to diffusion-based test sets). Therefore, I suggest that the authors also report results trained on a GAN-based dataset for these two benchmarks. This would ensure a fair comparison and further demonstrate that the proposed method is also effective when trained on GAN-generated data.

**Questions:**

I have no further questions.

---

> ### Author Response · Authors · 2025-11-27
> **Response to Reviewer EuiK (Part 1)**
>
> We sincerely thank Reviewer EuiK for the thoughtful and constructive feedback. The response is as follows.
>
> ---
>
> > *Q1: In the second paragraph of the introduction, the term “patch” appears for the first time but lacks a clear definition or motivation. Since AIGI detection includes many CNN-based detectors that do not explicitly rely on patch-level representations, introducing the patch concept without clarification may confuse readers about its relevance to this task.*
>
> **A:** We appreciate the reviewer's constructive comment regarding the definition and motivation of the "patch" concept.
>
> In our context, a "patch" refers to the fundamental input unit used in Vision Transformers (ViTs). Specifically, an input image is rasterized into a grid of fixed-size, non-overlapping sub-regions (e.g., $14 \times 14$ pixels), where each sub-region is treated as a discrete token and projected into a feature embedding.
>
> The emphasis on "patches" stems from the rapid evolution of the AIGI detection field. While early methods relied on CNNs, recent research has largely shifted toward ViT-based architectures [2–6] due to their superior generalization capabilities [1] compared to traditional CNNs. Since these dominant methods process images at the patch level, analyzing artifacts through the lens of "patches" has become essential for understanding and advancing modern AIGI detectors.
>
> We have revised the introduction (Line 43) to explicitly define "patches" as "local fixed-size sub-regions" to prevent confusion.
>
> ---
>
> > *Q2: In line 364, the reference of SAFE is wrong.*
>
> **A:** We thank the reviewer for identifying this error. We have corrected the citation for SAFE (line 364) in the revised manuscript.

---

> ### Author Response · Authors · 2025-11-27
> **Response to Reviewer EuiK (Part 2)**
>
> > *Q3: I suggest that the authors also report results trained on a GAN-based dataset for these two benchmarks. This would ensure a fair comparison and further demonstrate that the proposed method is also effective when trained on GAN-generated data.*
>
> **A:** We thank the reviewer for this suggestion regarding experimental fairness and domain generalization.
>
> To rigorously address the concern about domain-specific bias from Stable Diffusion reconstruction, we conducted a new experiment where the entire PPL framework was adapted to a GAN-based pipeline. Specifically, we trained on ProGAN (4-class subset: car, cat, chair, horse) and replaced the reconstruction module with VQGAN.
>
> As shown in Table 1 and Table 2, our method (PPL-VQGAN) consistently outperforms state-of-the-art baselines on both UniversalFakeDetect ($\uparrow$ 0.61%) and AIGCDetect ($\uparrow$ 2.45%), even when trained solely on GAN data. This confirms that the PPL framework is agnostic to the specific generative architecture used for reconstruction.
>
> **Table 1.** $\text{mACC}$ (%) comparison on UniversalFakeDetect. Results sourced from [3].
> |Method|ProGAN|CycleGAN|BigGAN|StyleGAN|GauGAN|StarGAN|Deepfakes|SITD|SAN|CRN|IMLE|Guided|LDM 200|LDM 200 cfg|LDM 100|GLIDE 100 27|Glide 50 27|Glide 100 10|DALLE|mAcc|
> |-|-|-|-|-|-|-|-|-|-|-|-|-|-|-|-|-|-|-|-|-|
> |UniFD|100.0| 98.50| 94.50| 82.00| 99.50| 97.00| 66.60| 63.00| 57.50| 59.5| 72.00| 70.03| 94.19| 73.76| 94.36| 79.07| 79.85| 78.14| 86.78| 81.38|
> |LGrad|97.90| 95.84| 90.45| 97.55| 90.24| 93.41| 97.40| 88.92| 59.04| 71.92| 67.35| 86.70| 84.55| 99.58| 65.56| 85.69| 97.40| 88.15| 59.06| 85.09|
> |FreqNet|99.84| 95.00| 87.55| 96.23| 86.57| 99.75| 76.89| 66.94| 98.63| 50.00| 50.00| 84.55| 97.65| 98.00| 98.20| 96.25| 97.15| 97.35| 87.15| 87.56|
> |FatFormer|99.89| 99.32| 99.50| 97.15| 99.41| 99.75| 93.23| 81.11| 68.04| 69.45| 69.45| 76.00| 98.60| 94.90| 98.65| 94.35| 94.65| 94.20| 98.75| 90.86|
> |C2P-CLIP|99.98| 97.31| 99.12| 96.44| 99.17| 99.60| 93.77| 95.56| 64.38| 93.29| 93.29| 69.10| 99.25| 97.25| 99.30| 95.25| 95.25| 96.10| 98.55| 93.79|
> |PPL-VQGAN|99.91| 98.18| 93.77| 97.64| 94.79| 99.42| 90.88| 75.56| 75.11| 95.6| 99.56| 85.2| 99.7| 99.25| 99.7| 94.75| 94.85| 95.4| 99.45| **94.40 ($\uparrow$ 0.61)** |
>
> **Table 2.** $\text{mACC}$ (%) comparison on UniversalFakeDetect. Results sourced from [4].
> |Method|ProGAN|StyleGAN|BigGAN|CycleGAN|StarGAN|GauGAN|StyleGAN2|WFR|ADM|Glide|Midjourney|SDv1.4|SDv1.5|VQDM|Wukong|DALLE2|mAcc
> |-|-|-|-|-|-|-|-|-|-|-|-|-|-|-|-|-|-|
> | CNNSpot| 100.00 | 90.17| 71.17| 87.62| 94.60| 81.42| 86.91| 91.65 | 60.39 | 58.07 | 51.39| 50.57| 50.53| 56.46 | 51.03| 50.45| 70.78|
> | UnivFD| 99.81| 84.93| 95.08| 98.33| 95.75| 99.47| 74.96| 86.90 | 66.87 | 62.46 | 56.13| 63.66| 63.49| 85.31| 70.93 | 50.75| 78.43|
> | DIRE-G| 95.19| 83.03| 70.12| 74.19| 95.47| 67.79| 75.31| 58.05 | 75.78 | 71.75 | 58.01| 49.74| 49.83| 53.68 | 54.46| 66.48| 68.68 |
> | DIRE-D| 52.75| 51.31| 49.70| 49.58| 46.72| 51.23| 51.72| 53.30 | 98.25 | 92.42 | 89.45| 91.24| 91.63| 91.90 | 90.90| 92.45| 71.53 |
> |NPR| 99.79| 97.70|84.35| 96.10|99.35|82.50| 98.38| 65.80 | 69.69 | 78.36 | 77.85| 78.63| 78.89| 78.13 | 76.11| 64.90|82.91|
> | PatchCraft | 100.00 | 92.77| 95.80| 70.17| 99.97| 71.58| 89.55| 85.80 | 82.17 | 83.79 | 90.12| 95.38| 95.30| 88.91 | 91.07| 96.60| 89.31 |
> |PPL-VQGAN|99.94|99.59|90.08|97.05|99.17|89.19|96.93|95.8|88.98|92.13|64.03|96.13|95.74|95.17|96.14|77.25|**91.76 ($\uparrow$ 2.45)**|
>
> ---
>
> [1] Towards Universal Fake Image Detectors that Generalize Across Generative Models, CVPR 2023.
>
> [2] DRCT: Diffusion Reconstruction Contrastive Training towards Universal Detection of Diffusion Generated Images, ICML 2024.
>
> [3] C2P-CLIP: Injecting Category Common Prompt in CLIP to Enhance Generalization in Deepfake Detection, AAAI 2025.
>
> [4] PatchCraft: Exploring Texture Patch for Efficient AI-generated Image Detection. Arxiv.
>
> [5] Forgery-aware Adaptive Transformer for Generalizable Synthetic Image Detection, CVPR 2024.
>
> [6] Effort: Orthogonal Subspace Decomposition for Generalizable AI-Generated Image Detection, ICML 2025.

---

### Official Review · Reviewer_LYB1 · 2025-10-31

**Soundness:** 3
**Presentation:** 3
**Contribution:** 2
**Rating:** 2
**Confidence:** 5

**Summary:**

This paper proposes two principles for AI-generated image (AIGI) detection: "All Patches Matter"  and "More Patches Better". The authors contend that existing detectors suffer from "Few-Patch Bias," relying excessively on a minimal number of highly discriminative patches. To address this issue, they propose the Panoptic Patch Learning (PPL) framework, comprising Randomized Patch Reconstruction (RPR) and Patch-wise Contrastive Learning (PCL). The method achieves superior cross-model generalization performance.

**Strengths:**

- The methodology is well-aligned with the motivation: RPR corresponds to the principle of "More Patches Better," while PCL reflects the idea that "All Patches Matter." Both methods are clearly motivated and logically consistent with the proposed principles.
- The experiments are comprehensive: performance is reported on benchmarks such as GenImage, DRCT-2M, and Chameleon, and robustness under various corruptions (e.g., compression and blur) is demonstrated. The study also includes ablation studies and hyperparameter analysis.

**Weaknesses:**

- The paper presents “All Patches Matter / More Patches Better” as a primary principle, but ideas such as “any local patch of an AI-generated image contains artifacts, and even a single patch can be sufficient for reliable discrimination” have already been explored in prior patch-based detection work discussed in the related work section. This makes the proposed patch-level principles feel incremental rather than conceptually new.
- Similarly, although the proposed PPL framework is shown to be effective, it is essentially an engineering-level refinement rather than a fundamentally new paradigm. The core idea of RPR is to move the DRCT-style diffusion reconstruction from the image level down to the patch level, while PCL is a straightforward application of supervised contrastive learning at the patch-token level.
- The paper relies on TDE for attribution. However, TDE does not appear to be a commonly adopted attribution method in deep learning. Why not use more standard interpretability approaches, such as Grad-CAM and its variants, or LRP (Layer-wise Relevance Propagation) [1], to explain what the detector is actually using?
- The paper focuses primarily on CLIP, DINOv2, and other ViT-style backbones, while largely ignoring CNN-based detectors. This raises an important question: do the core principles proposed in this paper still hold under CNN-based architectures?
- Going further, [2] uses Guided-GradCAM and LRP as attribution methods, extracts transferable forensic features from different layers of a CNN-based detector, and maps them back to the input image patches. The results indicate that color statistics are a key signal for CNN-based forgery detectors, rather than an extreme reliance on a few spatial patches. Different layers highlight different input regions. This behavior is not the same type of bias that the authors call “Few Patch Bias.” Therefore, the bias analysis in this paper may not generalize to CNNs, and it is not yet convincing that the claimed principles are architecture-agnostic.

[1] Transformer Interpretability Beyond Attention Visualization

[2] Discovering Transferable Forensic Features for CNN-generated Images Detection

**Questions:**

1. In Section 5, the citation of SAFE is incorrect.
2. In Table 6, the row labeled “Infonce/tau=0.5” is not aligned with the notation used in the main text.
3. In Section 3.1, the text uses “donot,” which is a typographical error. It should be “do not.”
4. In Algorithm 1, the classification loss is denoted as ( L_{ce} ) in Step 3, but it appears as ( L_{bce} ) in Step 5 when forming the total loss.
5. The terminology for the proposed bias is inconsistent. The Introduction (in the italicized part) uses “Few Patch Bias,” while other parts of the paper use “Few-Patch Bias.”

---

> ### Author Response · Authors · 2025-11-27
> **Response to Reviewer LYB1 (Part 1)**
>
> We sincerely thank Reviewer LYB1 for the thoughtful and constructive feedback. The response is as follows.
>
> ---
>
> >*Q1: The paper presents “All Patches Matter / More Patches Better” as a primary principle, but ideas such as “any local patch of an AI-generated image contains artifacts, and even a single patch can be sufficient for reliable discrimination” have already been explored in prior patch-based detection work discussed in the related work section. This makes the proposed patch-level principles feel incremental rather than conceptually new.*
>
> **A:** We appreciate the opportunity to clarify the distinctions between our work and prior patch-based detection methods. While we acknowledge that previous studies (e.g., SSP [1], PatchCraft [2]) have explored local artifacts, our framework is driven by a fundamentally different design philosophy: moving from using a single or selected patches to using all patches. We detail the key conceptual differences below:
>
> 1. Contrast with SSP [1]: SSP relies on the premise that "A single simplest patch is capable of generalizing well". Our approach diverges from this by positing that "All Patches Matter," arguing that relying on a single patch risks overfitting and ignoring complex, distributed artifacts. By leveraging the full spectrum of available patches, our framework achieves significantly better generalizability. As shown in Table 1 and Table 2, this holistic strategy yields performance gains of **6.6%** (Table 1) and **12.89%** (Table 2), validating the effectiveness of utilizing richer artifact cues.
>
> 2. Contrast with PatchCraft [2]: PatchCraft employs a selective sampling strategy, effectively discarding a portion of the input (using only $1/3$ richest and $1/3$ simplest textures). We argue that this approach inadvertently filters out potential forensic cues present in the discarded regions. Conversely, our method actively utilizes all available patches, ensuring no discriminative information is lost. This comprehensive coverage results in superior detection performance, evidenced by improvements of **14.9%** (Table 1) and **16.37%** (Table 2) over the PatchCraft baseline.
>
> **Table 1.** $\text{mACC}$ (%) comparison on GenImage. Results sourced from [1] and [2].
> |Methods|SDv1.4|SDv1.5|Midjourney|ADM|Wukong|GLIDE|VQDM|BigGAN|mACC(%)|
> |-|-|-|-|-|-|-|-|-|-|
> |SSP|99.2|99.3|82.6|78.9|88.9|98.6|96.0|73.9|90.6 **($\downarrow$ 6.6)**|
> |PatchCraft|89.5|89.3|79.0|77.3|78.4|89.3|83.7|72.4|82.3 **($\downarrow$ 14.9)**|
> |PPL|98.5|98.3|94.8|94.7|98.6|96.1|98.5|98.0|**97.2**|
>
> **Table 2.** $\text{mACC}$ (%) comparison against baselines. PatchCraft results sourced from [2]; SSP results from our implementation.
> |Methods|SSP|PatchCraft|PPL|
> |-|-|-|-|
> |mACC(%)|59.18 **($\downarrow$ 12.89)**|55.70 **($\downarrow$ 16.37)**|**72.07**|

---

> ### Author Response · Authors · 2025-11-27
> **Response to Reviewer LYB1 (Part 2)**
>
> >*Q2: Similarly, although the proposed PPL framework is shown to be effective, it is essentially an engineering-level refinement rather than a fundamentally new paradigm. The core idea of RPR is to move the DRCT-style diffusion reconstruction from the image level down to the patch level, while PCL is a straightforward application of supervised contrastive learning at the patch-token level.*
>
> **A:** We respectfully disagree with the characterization of PPL as incremental engineering. The novelty lies in the synergistic integration of these components to solve a specific, pervasive failure mode in detection:"Few-Patch Bias" (over-reliance on dominant local artifacts).
>
> The framework is not merely applying global methods locally; it is a designed counter-measure mechanism:
>
> * **Breaking Shortcuts (RPR):** Unlike global DRCT, RPR specifically targets and reconstructs local patches to disrupt dominant artifacts, forcing the model to abandon "shortcuts" and explore harder regions.
>
> * **Enforcing Uniformity (PCL):** PCL ensures that these newly explored, non-dominant regions contribute equally to the decision boundary.
>
> **Validation of Synergy:** The distinction between "engineering tweaks" and a "cohesive framework" is mathematically evident in our ablation study (Table 3): While the individual application of PCL **($\uparrow$ 0.9%)** or RPR **($\uparrow$ 1.4%)** offers modest improvement, their integration of both components delivers a substantial performance gain **($\uparrow$ 6.0%)**. This non-linear synergistic effect proves that RPR and PCL are mutually amplifying components indispensable for the holistic success of the PPL paradigm.
>
> **Table 3.** Ablation experiment of each components on GenImage.
> |Methods|mACC(%)|
> |-|-|
> |CLIP|91.2|
> |CLIP+PCL|92.1 ($\uparrow$ 0.9)|
> |CLIP+RPR|92.6 ($\uparrow$ 1.4)|
> |CLIP+RPR+PCL|**97.2 ($\uparrow$ 6.0)**|

---

> ### Author Response · Authors · 2025-11-27
> **Response to Reviewer LYB1 (Part 3)**
>
> >*Q3: The paper relies on TDE for attribution. However, TDE does not appear to be a commonly adopted attribution method in deep learning. Why not use more standard interpretability approaches, such as Grad-CAM and its variants, or LRP (Layer-wise Relevance Propagation) [4], to explain what the detector is actually using?*
>
> **A:** We appreciate the reviewer’s suggestion to incorporate standard interpretability methods to validate our analysis. We have included **Figure 17 in Appendix C (page 18)** to visualize Grad-CAM as you recommended.
>
> **Qualitative Analysis (Grad-CAM Visualization):** Following the official implementation [5], we computed gradients with respect to the feature maps of the final Layer Normalization layer to generate activation heatmaps.
>
> * **Baseline:** As shown in Figure 17, the baseline (naively LoRA-tuned) model exhibits "Few-Patch Bias," focusing on sparse, isolated regions.
> * **Ours (PPL):** In contrast, PPL demonstrates a significantly broader and more spatially uniform distribution, corroborating our earlier TDEd findings that our framework effectively forces the model to leverage distributed artifacts.
>
> **Quantitative Analysis (Coverage & Entropy):** To provide a rigorous comparison, we introduced two metrics to evaluate the heatmap distributions:
>
> * **Coverage:** The proportion of pixels with heatmap intensity $> \tau$ ($\tau =0.1$).
> $$\text{Coverage} = \frac{1}{H \times W} \sum_{i=1}^{H} \sum_{j=1}^{W} \mathbb{1}(M_{i,j} > \tau)$$
> where $\mathbb{1}(\cdot)$ is the indicator function, which equals 1 if the condition holds and 0 otherwise.
>
> * **Entropy:** Measures the dispersion of activations (indicative of global feature usage).
> $$\text{Entropy} = - \sum_{i=1}^{H} \sum_{j=1}^{W} P_{i,j} \log(P_{i,j} + \epsilon), \quad \text{where } P_{i,j} = \frac{M_{i,j}}{\sum_{x,y} M_{x,y}}$$
> where $P_{i,j} = \frac{M_{i,j}}{\sum_{x,y} M_{x,y}}$ represents the normalized heatmap intensity at position $(i,j)$, and $\epsilon$ is a small constant for numerical stability.
>
> **Results:** As detailed in Table 4, PPL achieves substantially higher Coverage (43.6% vs 26.9%) and Entropy (9.16 vs 8.70). These metrics quantify that PPL successfully mitigates over-reliance on local shortcuts by utilizing global information.
>
> **Table 4.** Quantitative statistics of Grad-CAM heatmaps comparing Baseline (CLIP) and PPL.
> |Methods|Coverage $\uparrow$|Entropy$\uparrow$|
> |-|-|-|
> |CLIP|26.9|8.70|
> |CLIP+PPL|**43.6%**|**9.16**|
>
> ---
>
> >*Q4: The paper focuses primarily on CLIP, DINOv2, and other ViT-style backbones, while largely ignoring CNN-based detectors. This raises an important question: do the core principles proposed in this paper still hold under CNN-based architectures?*
>
> **A:** We appreciate the reviewer for raising the crucial question of architectural generalization. We confirm that our principles hold for CNN-based detectors through both visual diagnosis (existing in paper) and quantitative mitigation (new experiments).
>
> To validate this, we applied the Random Patch Reconstruction (RPR) module—which fulfills the "More Patches Better" principle, on ConvNeXt [6], a state-of-the-art CNN architecture and trained on the SDv1.4 subset of the DRCT dataset.
>
> As detailed in Table 5, the integration of RPR results in consistent performance improvements, demonstrating that our framework is model-agnostic and effective across architectures.
>
> **Table 5.** $\text{mACC}$ (%) comparison on DRCT. Results sourced from DRCT[7].
> | Methods | LDM | SDv1.4 | SDv1.5 | SDv2 | SDXL | SDXLRefiner | SDTurbo | SDXLTurbo | LCMSDv1.5 | LCMSDXL | SDv1-Ctrl | SDv2-Ctrl | SDXL-Ctrl | SDv1-DR | SDv2-DR | SDXL-DR | mAcc |
> | - | - | - | - | - | - | - | - | - | - | - | - | - | - | - | - | - | - |
> |CNNSpot|99.87|99.91|99.90|97.55|66.25|86.55|86.15|72.42|98.26|61.72|97.96|85.89|82.84|60.93|51.41|50.28|81.12|
> |F3Net|99.85|99.78|99.79|88.66|55.85|87.37|68.29|63.66|97.39|54.98|97.98|72.39|81.99|65.42|50.39|50.27|77.13|
> |CLIP/RN50|99.00|99.99|99.96|94.61|62.08|91.43|83.57|64.40|98.97|57.43|99.74|80.69|82.03|65.83|50.67|50.47|80.05|
> |GramNet|99.40|99.01|98.84|95.30|62.63|80.68|71.19|69.32|93.05|57.02|89.97|75.55|82.68|51.23|50.01|50.08|76.62|
> |De-fake|98.30|96.22|96.33|93.83|91.01|93.91|86.38|85.92|90.44|88.99|90.41|81.06|89.06|51.96|51.03|50.46|83.46|
> |UnivFD|99.97|100.0|99.97|95.84|64.44|82.00|80.82|60.75|99.27|62.33|99.80|83.40|73.28|61.65|51.79|50.41|79.11|
> |ConvNext-Base|92.1|99.53|99.51|89.65|64.02|69.24|92.00|93.93|99.13|70.89|58.98|62.34|66.66|50.12|50.16|50.00|75.52|
> |ConvNeXt DRCT| 99.66|98.56|98.48|99.85|96.10|98.68|99.59|83.30|98.45|93.78|96.68|99.85|97.66|93.91|99.87|90.39|96.55|
> |ConvNeXt + RPR| 98.81|98.81|98.77|98.72|98.41|98.76|98.69|98.42|98.77|98.14|98.81|98.69|98.51|97.14|98.69|97.98|**98.51 ($\uparrow$ 1.96)**|

---

> ### Author Response · Authors · 2025-11-27
> **Response to Reviewer LYB1 (Part 4)**
>
> >*Q5: Going further, [5] uses Guided-GradCAM and LRP as attribution methods, extracts transferable forensic features from different layers of a CNN-based detector, and maps them back to the input image patches. The results indicate that color statistics are a key signal for CNN-based forgery detectors, rather than an extreme reliance on a few spatial patches. Different layers highlight different input regions. This behavior is not the same type of bias that the authors call “Few Patch Bias.” Therefore, the bias analysis in this paper may not generalize to CNNs, and it is not yet convincing that the claimed principles are architecture-agnostic.*
>
> **A:** We thank the reviewer for this insightful comment regarding the generalizability of our principles to CNN architectures. We agree that shallow convolutional layers focus on varied local cues as they extract preliminary features, and deeper layers aggregate information for high-level decision-making, is a known property of CNNs [9].
>
> In Figure 18 (Appendix C, page 19), we visualize the activation maps of the deepest **3** blocks of a ConvNeXt [7] detector trained on GenImage using the official Grad-CAM [6] implementation. These visualizations clearly demonstrate that in the final decision-making stages, the CNN's attention converges onto a limited set of dominant regions, mirroring the "Few Patch Bias" observed in ViTs.
>
> ---
>
> [1] A Single Simple Patch is All You Need for AI-generated Image Detection. Arxiv.
>
> [2] PatchCraft: Exploring Texture Patch for Efficient AI-generated Image Detection. Arxiv.
>
> [3] A Sanity Check for AI-generated Image Detection. ICLR 2025.
>
> [4] Discovering Transferable Forensic Features for CNN-generated Images Detection. ECCV 2022.
>
> [5] Grad-CAM: Visual Explanations from Deep Networks via Gradient-based Localization. ICCV 2017.
>
> [6] A ConvNet for the 2020s. CVPR 2022.
>
> [7] DRCT: Diffusion Reconstruction Contrastive Training towards Universal Detection of Diffusion Generated Images. ICML 2024.
>
> [8] Transformer Interpretability Beyond Attention Visualization. CVPR 2021.
>
> [9] Visualizing and Understanding Convolutional Networks. ECCV 2014.

---

### Author Response · Authors · 2025-12-03
**Summary of discussion during the rebuttal period**

**Dear Area Chair**,

Thank you very much for serving as the Area Chair for our paper. We value this opportunity to provide a concise summary to assist in your decision-making. We are encouraged that the reviewers unanimously found the **insightful motivation** (`Reviewers EuiK`, `sAHA`, and `JM5D`), the **SOTA performance** (`Reviewers EuiK` and `JM5D`), the **extensive experiments** (`Reviewers LYB1`, `EuiK`, and `sAHA`).

For your convenience, we have summarized our major rebuttal responses as follows:

---

**Response to `Reviewer LYB1` (Initial Score: 2)**

* **Concerns regarding the conceptual novelty of the proposed patch-level principles:**
    * Unlike SSP and PatchCraft's selective strategies, our emphasis on utilizing information from all patches significantly enhances generalization, yielding substantial performance gains of **6.6%–12.9%** and **14.9%–16.4%** respectively. (Q1 Table 1&2.)

* **Concerns regarding PPL being merely incremental engineering of DRCT and contrastive learning：**
    * PPL represents a synergistic framework rather than incremental engineering. Empirically, while individual modules yield minor gains **(0.9%/1.4%)**, their combination delivers a non-linear **6.0%** boost, validating mutual amplification. (Q2 Table 3.)

* **Questioning the reliance on TDE over standard attribution methods like Grad-CAM.**
    * Grad-CAM analysis confirms PPL mitigates local bias by activating broader regions, quantitatively evidenced by significantly higher Coverage (**43.6%** vs 26.9%) and Entropy (**9.16** vs 8.70). (Q3 Table 4.)

* **Questioning the generalizability of proposed principles to CNN-based architectures:**
    * Our principles generalize effectively to CNN-based architectures. Integrating RPR into ConvNeXt yields consistent improvements, achieving **98.51%** mACC—a **1.96%** gain over the DRCT baseline. (Q4 Table 5.)

* **Questioning the applicability of Few-Patch Bias to CNN architectures:**
    * "Few-Patch Bias" inherently exists in CNNs' deep decision-making layers. Grad-CAM visualizations of ConvNeXt's final blocks (Appendix Figure 18) confirm attention converges on limited dominant regions, mirroring ViT behavior. (Q5 Table 6.)

---

**Response to `Reviewer sAHA` (Initial Score: 6)**

* **Questioning the justification and necessity of the All Patches Matter principle:**
    * "All Patches Matter" denotes pervasive artifacts originating from upsampling. While single-patch detection is viable (**89.6%** mACC), our holistic PPL framework significantly enhances generalization, achieving **97.2%** mACC (**$\uparrow$ 7.6%**). (Q1-4 Table 1&2.)

* **Questioning if attention bias arises from ViT register tokens:**
    * Experiments confirm register tokens yield only marginal gains (**$\uparrow$ 1.0%**) and fail to solve the bias. Conversely, PPL provides a superior solution, significantly boosting DINOv2 accuracy to 95.9%. (Q5 Table 3.)

* **Requesting comparison with prior reconstruction-based methods *Aligned* and *Bias-Free*:**
    * Unlike global reconstruction methods, our randomized patch injection effectively captures fine-grained artifacts. PPL outperforms *Aligned* by **14.2%** and *Bias-Free* by **6.0%**. (Q8 Table 4&5.)

* **Questioning the feasibility of training solely with real and reconstructed images:**
    * Reconstruction-based artifacts alone are sufficient for robust feature learning. Training exclusively on real and RPR images yields comparable results (**94.2%** vs **97.2%**), validating the efficacy of generated cues. (Q11 Table 6)

---

**Response to `Reviewer JM5D` (Initial Score: 6; Updated to 8)**

* **Questioning RPR's domain bias and requesting GAN-based validation for generality:**
    * To address domain bias, we adapted PPL to a GAN-based pipeline (training on ProGAN with VQGAN reconstruction). PPL-VQGAN achieves SOTA on *UniversalFakeDetect* (mACC $\uparrow$ **0.61%**) and *AIGCDetect* (mACC $\uparrow$ **2.45%**). (Q1 Table 1&2)

* **Questioning the stability of the Dropout ablation and the validity of PPL's advantage:**
    * Repeated experiments confirm the Dropout ablation is highly unstable due to stochasticity. In contrast, PPL demonstrates superior robustness, achieving the highest mean accuracy of **96.9%** with significantly low standard deviation (**$\pm 0.64$**). (Q2 Table 3)

---

**Response to `Reviewer EuiK` (Initial Score: 8)**

* **Suggesting evaluation of GAN-trained models to ensure fair comparison and validate framework generality:**
    * To validate generality, we adapted PPL to a GAN-based pipeline (PPL-VQGAN). This adaptation consistently outperforms baselines, achieving performance gains of 0.61% on UniversalFakeDetect and 2.45% on AIGCDetect. (A3 Table 1&2)

---

Thank you again for your efforts in handling our submission.

Best regards,

**Authors of Paper #10301**

---

### Meta-Review · Area_Chair_obNR · 2026-01-07

**Summary:**

This paper received mixed initial review comments. Most of the reviews are positive, mainly favoring the motivation of the paper, and the simple yet effective solution on improving the information usage efficiency for decision-making. The visualization analysis and the solution inspired by the analysis is well coupled and clearly explained. One of the reviewer argue about the novelty and the engineering-level of the paper. AC reads both the review comments and the rebuttal, and believes that compared with previous works, the novel motivation and the patch-level contrastive learning is significant enough. The performance gains demonstrated in metrics and visuals are convincing. AC recommends to accept the paper as a poster presentation. More details are below.

**Reviewer Concerns:**

- Reviewer LYB1: the reviewer has concerns on the novelty of the motivation that any patch contains rich information for detecting artifacts. The authors made a great point that using more patches effectively can better enhance the detection accuracy and demonstrated it by comparing with previous patch-based methods. AC is convinced that the difference of the idea is significant. The reviewer also pointed out that the proposed method is more like an engineering improvements over existing approach. AC still believes the combination of the proposed data augmentation and the training strategy is non-trivial, and the two components are well coupled. Regarding CNN-based detector, the authors also show similar findings via metrics and visualization, which is convincing enough. However, the reviewer might not agree with the authors because the overall proposal is combining and extending previous works from image-level to patch-level. AC understands the concerns and feels it not a major blocker for acceptance.
- Reviewer EuiK: the reviewer does not have major concerns, but mentioned potentially missing experiments on difference domains of the training data. The authors made up the experiments in a convincing way, demonstrating the solid conclusion of the generalization capability.
- Reviewer sAHA: the reviewer mentions the unclear statement in the motivation. The authors further clarified that effectively utilizing more patches is better than using a single patch, or feeding in more patches directly. AC encourages the authors to revise some statements to resolve potential confusion. Also AC suggests the authors to include some discussions on partially-edited images, but not limited the method to full image. In this cases, it will be interesting to analysis whether all-patch-matters claim still stand. Other concerns are addressed well.
- Reviewer JM5D: multiple experiments are made up to fully address the reviewer's concerns.

**Reviewer Scores:**

- Reviewer LYB1: it is not easy for the reviewer to change the score given the research taste towards higher quality scientific analysis and fundamental research.
- Reviewer EuiK: it is possible to improve the score, or keep it unchanged.
- Reviewer sAHA: it is possible to improve the score, or keep it unchanged.
- Reviewer sAHA: it is possible to improve the score.

---

### Decision · Program_Chairs · 2026-01-26

Accept (Poster)